# Small molecule in situ resin capture provides a compound first approach to natural product discovery

Alexander Bogdanov[1], Mariam N. Salib[2], Alexander B. Chase [1,3], Heinz Hammerlindl [4], Mitchell N. Muskat [1], Stephanie Luedtke [5], Elany Barbosa da Silva[5], Anthony J. O'Donoghue[5], Lani F. Wu [4], Steven J. Altschuler [4], Tadeusz F. Molinski [2] ✉ & Paul R. Jensen [1] ✉

Culture-based microbial natural product discovery strategies fail to realize the extraordinary biosynthetic potential detected across earth's microbiomes. Here we introduce Small Molecule In situ Resin Capture (SMIRC), a culture-independent method to obtain natural products directly from the environments in which they are produced. We use SMIRC to capture numerous compounds including two new carbon skeletons that were characterized using NMR and contain structural features that are, to the best of our knowledge, unprecedented among natural products. Applications across diverse marine habitats reveal biome-specific metabolomic signatures and levels of chemical diversity in concordance with sequence-based predictions. Expanded deployments, in situ cultivation, and metagenomics facilitate compound discovery, enhance yields, and link compounds to candidate producing organisms, although microbial community complexity creates challenges for the later. This compound-first approach to natural product discovery provides access to poorly explored chemical space and has implications for drug discovery and the detection of chemically mediated biotic interactions.

Microbial natural products account for many of today's essential medicines, including most antibiotics[1,2]. These compounds are traditionally discovered using a 'microbe-first' approach, where individual strains are cultured in the laboratory and bioassays used to guide the isolation of active compounds from culture extracts. While new compounds are regularly discovered using this approach, it has become increasingly difficult to find fundamentally new chemical scaffolds, a critical resource for drug discovery and a driving force behind subsequent synthetic and biosynthetic studies. Recent analyses of genomic, metagenomic, and amplicon data all indicate the existence of extensive and yet to be realized natural product biosynthetic diversity across Earth's microbiomes[3–7]. Despite this realization, our best discovery efforts have failed to access the vast majority of predicted chemical space, and thus new approaches to natural product discovery are needed.

The challenges associated with microbial natural product discovery are well documented[8] and include the re-isolation of known compounds[9], the recognition that only a small percentage of bacterial diversity has been cultured[10], and observations that most of the genes associated with natural product biosynthesis are silent under laboratory growth conditions[11]. Efforts to address these challenges include improved cultivation methods[12], co-cultivation or the addition of elicitors to activate silent biosynthetic gene clusters (BGCs)[13,14], and genome mining[15,16], where BGCs are activated via genetic

[1]Scripps Institution of Oceanography, University of California, San Diego, La Jolla, CA 92093, USA. [2]Department of Chemistry and Biochemistry, University of California, San Diego, La Jolla, CA 92093, USA. [3]Department of Earth Sciences, Southern Methodist University, Dallas, TX 75275, USA. [4]Department of Pharmaceutical Chemistry, University of California, San Francisco, San Francisco, CA 94158, USA. [5]Skaggs School of Pharmacy and Pharmaceutical Sciences, University of California, San Diego, La Jolla, CA 92093, USA. ✉e-mail: tmolinski@ucsd.edu; pjensen@ucsd.edu

manipulation[17] or heterologous expression[18]. While these methods have shown promise, they remain time consuming, are primarily limited to the small percentage of microbial diversity that can be cultured[19], and fail to capture the rich natural product diversity predicted to be encoded by the Earth's collective microbiome, which is estimated to encompass as many as one trillion microbial species[20].

Here we report a new approach to microbial natural product discovery that bypasses the need for laboratory cultivation. Rather than relying on cultured strains to drive discoveries, compounds are captured directly from the environments in which they are produced using an adsorbent resin without consideration of the producing organism. This culture-independent approach, which we call Small Molecule In Situ Resin Capture (SMIRC), builds upon solid-phase methods developed to monitor marine toxins[21], detect known microbial products in marine sediments[22], probe the biosynthetic origins of marine natural products[23], and detect soft coral[24] and sponge metabolites in seawater[25]. Unlike traditional discovery approaches, SMIRC can capture novel natural products originating from bacteria to phytoplankton in a single deployment and, as we show, some that were previously reported from marine plants and invertebrates. As such, it requires no knowledge of cultivation conditions, or the factors required to induce biosynthetic gene expression, two significant obstacles to current discovery strategies. SMIRC was further modified to enhance in situ microbial growth with the aim of capitalizing on nature to provide the cues needed to trigger natural product biosynthesis. We demonstrate the potential of SMIRC for natural product discovery with the isolation and characterization of two new carbon skeletons, one of which possesses biological activity, and the detection of ten other new compounds, all from a single deployment site.

## Results

### SMIRC deployments and compound dereplication

Microbial natural products are generally secreted into the environment where they play important ecological roles as allelochemicals, siderophores, signaling molecules, and more[26–28]. SMIRC employs the adsorbent resin HP-20, which has proven effective for the adsorption of marine toxins across a wide range of polarities[29], to capture these small organic molecules after they are released and thus provides a mechanism to access environmental metabolomes. This technique avoids the extraction of media and intracellular components associated with traditional culture-based approaches, which can mask low-concentration compounds of interest. We first tested SMIRC in a *Zostera marina* seagrass meadow (<2 m depth) in Mission Bay, San Diego (Fig. 1A) and obtained salt-free, dark brown extracts averaging $54.2 \pm 6.1$ mg per replicate ($n = 4$). A microfractionation antibiotic assay targeting the outer membrane deficient *E. coli* strain lptD4123[30] revealed a large active peak with a UV maximum at 360 nm (Fig. 1B).

Major compounds in the active fraction ($m/z$ 381.0906 and 301.1396) were detected using high-resolution mass spectrometry (MS) with the mass difference of 79.951 Da attributed to the loss of $SO_3$ (calcd. 79.9568 Da). Bioassay-guided $C_{18}$ fractionation from one SMIRC replicate extract followed by HPLC purification yielded 2 mg of the major compound ($m/z$ 301.1396), which was identified as the flavonoid chrysoeriol based on the comparison of MS and NMR data with published reports (Fig. 1C). Sulfated flavonoids with antibacterial activity have been reported from *Z. marina*[31], providing a candidate producer for these compounds. While our goal was to capture novel microbial products, this first deployment demonstrated that compound yields sufficient for NMR-based structure elucidation could be obtained using SMIRC. In addition, it supports the use of resin capture techniques to address questions in chemical ecology[25], in this case the impact of seagrass metabolites on microbial community structure. The effectiveness of HP-20 to concentrate compounds from seawater was also evident as the 2 mg chrysoeriol captured using 100 g resin was

equivalent to the yields calculated from >1000 L of seawater (<1 mg/L chrysoeriol).

In a follow-up deployment at the same site, the SMIRC technique was modified by embedding the resin in an agar matrix with the aim of increasing the yields of microbial-derived compounds by facilitating in situ microbial growth. The addition of agar led to a different chemical profile and the detection of a major new peak in the UV chromatogram (Fig. 1D), which we isolated (0.4 mg) and identified as the α-pyrone aplysiopsene A (Fig. 1E) using NMR and tandem MS/MS (Supplementary Information). Aplysiopsene A was initially reported from the sacoglossan mollusk *Aplysiopsis formosa*[32] while structurally related compounds have been reported from a sea fan-derived fungus[33] and the actinobacterium *Nocardiopsis dassonvillei*[34]. Given that sacoglossans were not observed at the deployment site, the isolation of aplysiopsene A suggests that it may be of microbial origin and that SMIRC can provide new insights into the roles of microbes in the production of natural products reported from marine invertebrates[35,36]. Of note, we observed pink bacterial-like colonies (Fig. 1F) on the agar-resin matrix, indicating that this approach can support in situ cultivation and that the extract chemical profiles may be influenced by microbes growing on the agar surface. While efforts to cultivate aplysiopsene A producing bacteria from these samples have yet to be successful, these results provide a second example in which SMIRC deployments yielded sufficient compound for NMR-based structural elucidation and evidence that embedding the resin in agar can enhance compound discovery.

We next deployed SMIRC in the littoral zone at Cabrillo State Marine Reserve (CSMR, Fig. 2A), a relatively pristine habitat comprised of sand patches and rocky reef dominated by the seagrass *Phyllospadix* sp. and inhabited by a diverse assemblage of macroalgae and invertebrates.

Deployments (without agar) spanned 2–8 days depending on tides and swells, yielding extract masses averaging 177.1 mg (±87.6 mg, $n = 17$), with some exceeding 300 mg per 100 g resin. While no antibiotic activity was detected in the initial screen, LCMS analyses revealed extensive chemical complexity, visualized as congested peaks in the LC (UV, 254 nm) and MS (total ion count) chromatograms (Supplementary Fig. 1). This level of complexity was common to the crude extracts, which required extensive fractionation. The large number of molecules that could not be identified in these extracts suggested a high degree of chemical novelty and opportunities for natural product discovery guided by the unusual masses observed in the LCMS data.

### Natural product discovery

We first targeted a potentially novel compound from the CSMR extracts that had a relatively high-intensity molecular peak ($m/z$ 328.2485 [M + H]$^+$ Fig. 2B), suggesting it could be isolated. The calculated molecular formula of $C_{18}H_{33}NO_4$ (calcd [M + H]$^+$ 328.2483, 0.61 ppm) with three degrees of unsaturation matched three compounds in the Dictionary of Natural Products (https://dnp.chemnetbase.com): 3-hydroxy-$C_{14}$-homoserine lactone and curvularides B and E, initially reported from various bacteria and an endophytic fungus[37], respectively. We subjected two extracts with the highest estimated abundance of the compound to HPLC purification guided by UV and evaporative light-scattering (ELSD) detection given the absence of a strong chromophore. Repeated rounds of HPLC purification, supported by LCMS analyses, yielded ~40 μg of a pure compound as a white solid. The $^1H$ NMR spectrum (600 MHz, CD$_3$OD, 1.7 mm microcryoprobe) revealed three doublet resonances attributable to three methyl groups ($\delta_H$ 0.90, $J = 6.5$ Hz; 0.96, $J = 6.7$; 1.20, $J = 6.3$ Hz) positioned on methine carbons (Fig. 2C), thus indicating that our targeted compound is neither 3-hydroxy-$C_{14}$-homoserine lactone, which possesses only one methyl group, nor one of the curvularides, which have five methyl groups. The complete structure

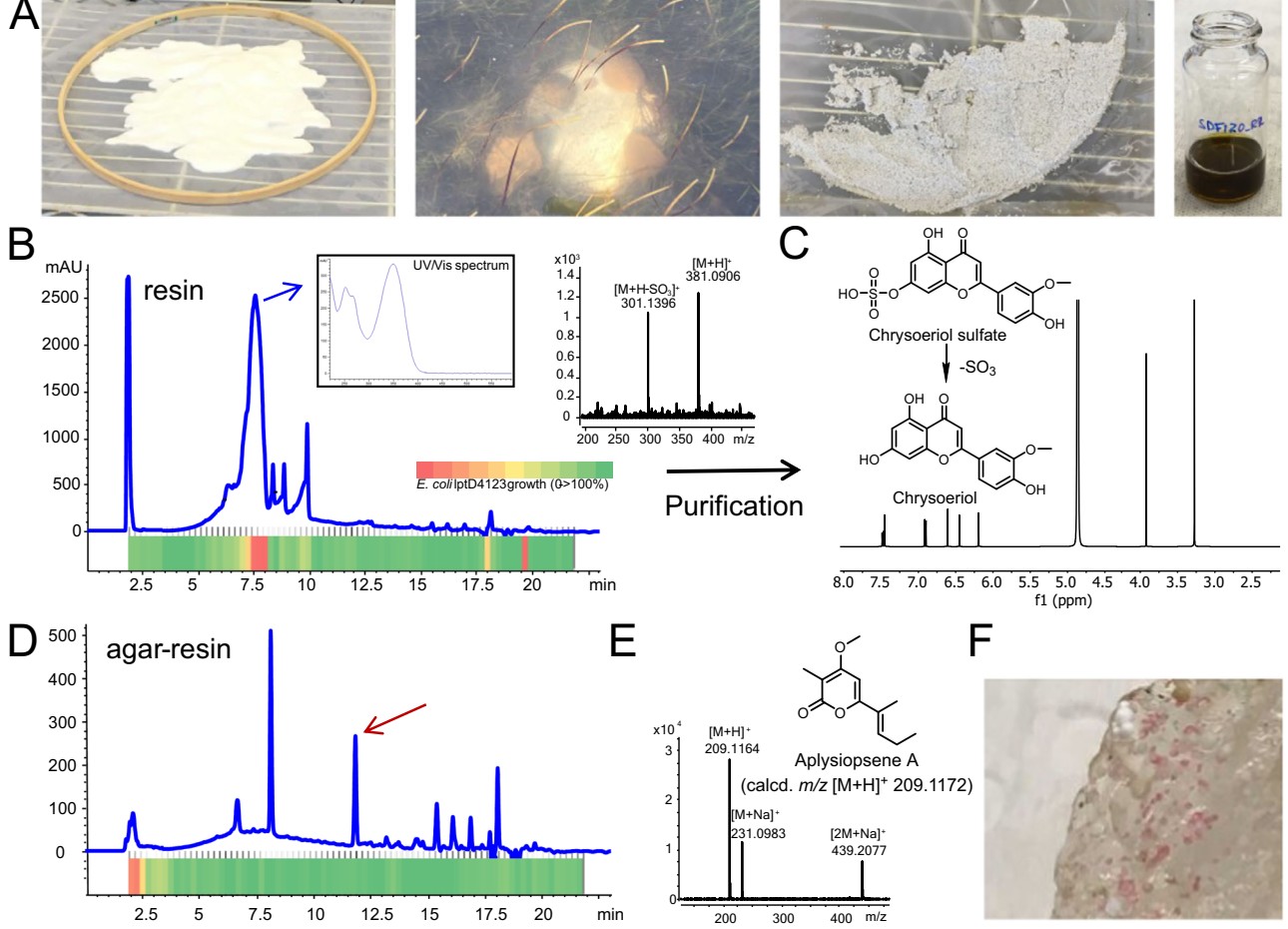

**Fig. 1 | SMIRC deployment in a *Zostera marina* meadow (Mission Bay, San Diego), extract analysis, compound identification, and in situ microbial growth. A** From left to right: activated HP-20 resin before enclosing in Nitex mesh and embroidery hoop, SMIRC deployment, recovered resin before extraction, concentrated extract in MeOH. **B** $UV_{360}$ chromatogram of the resin extract and corresponding microfractionation bioassay (80 fractions collected over 20 min) with heatmap showing *E. coli* LptD4123 growth inhibition ($OD_{650}$, red = no growth, green = 100% growth relative to control). Blue arrow: UV spectrum of active peak

(7.6 min). MS spectrum of the active compound shows loss of sulfate. **C** $^1H$ NMR (500 MHz) and structure of the flavonoid chrysoeriol, a degradation product of chrysoeriol sulfate. **D** $UV_{360}$ chromatogram of extract generated from resin embedded in agar (in situ cultivation) deployed at the same site revealed an additional peak (red arrow). **E** Structure and MS spectrum of aplysiopsene A isolated from the agar/resin peak (red arrow). **F** Pink colonies growing on agar/resin matrix. UV (ultraviolet), calcd (calculated), OD (optical density).

elucidation of this compound (Supplementary Notes, Supplementary Table 1, Supplementary Figs. 2–4), which we have called cabrillostatin (**1**, Fig. 3), was accomplished with 2D NMR experiments ($^1H$-$^1H$ COSY, HSQC, HMBC). Cabrillostatin represents a new carbon skeleton comprised of the uncommon, non-proteinogenic gamma-amino acid statine *N*-acylated to a 9-hydroxydecanoate ester to form a 15-membered macrocycle. The characterization of this compound by NMR validates the application of SMIRC for natural product discovery.

While isolating cabrillostatin (**1**), a second compound of interest was detected that displayed an isotopic pattern in the MS spectrum indicative of a dihalogenated molecule bearing one Cl and one Br (Fig. 2B). The molecular formula $C_{29}H_{40}BrClO_9$, with nine degrees of unsaturation, was calculated from the highest intensity isotopologue (Supplementary Fig. 5) at $m/z$ 629.1510 (calcd for $m/z$ [M-$H_2O$ + H]$^+$ 629.1512, 0.32 ppm). Database searches returned no matches for this formula suggesting it represented a new natural product. Multiple rounds of HPLC purification (Methods) yielded 50 µg of pure compound from a resin extract (100 mg). A panel of NMR experiments ($^1H$, $^1H$-$^1H$ COSY, HSQC, HMBC, NOESY, 600 MHz, 1.7 mm microcryoprobe, $CD_3CN$) conducted over long acquisition times (>72 h for HMBC) allowed assembly of most of the molecular structure but left several important substructures undefined due to missing H-C correlations in

the HMBC spectrum. Pooling samples from two SMIRC extracts yielded ~100 µg of compound that was subjected to NMR experiments ($^1H$, COSY, and HSQC), repeated in $CD_3OD$, that improved line shapes, dispersion of signals and *S/N*. An LR-HSQMBC (optimized for *J* = 8 Hz) experiment revealed additional correlations not observed in $CD_3CN$[38]. Extensive analysis of the NMR data recorded in both solvents and MS/MS fragmentation patterns (Supplementary Tables 2 and 3, Supplementary Figs. 6 and 7A) allowed us to establish the structure as a new bromo-chloro-polyketide that we have named cabrillospiral A (**2**) (Fig. 3).

The structure of cabrillospiral A (**2**) exhibits several rare features including an α-halogenated bicyclic [6,5]-spiroketal and an α-β unsaturated aldehyde appended to a C=C double bond that is exocyclic to a tetrahydrofuran. The latter, a vinylogous formate ester, is to the best of our knowledge, unprecedented among compounds in both the natural product and synthetic literature. Cabrillospiral A (**2**) has 11 chiral centers distributed across four independent stereogenic substructures, making stereochemical elucidation a challenge. Interpretation of the NOESY data, $^1H$-homonuclear scalar coupling ($^{2,3}J$), electronic circular dichroism (ECD) experiments, and DFT calculated ECD spectra led to the assignment of the $\Delta^2$ double bond geometry, the absolute configuration at the ring A as 5 *S*, 6 *S* and the relative configuration of four

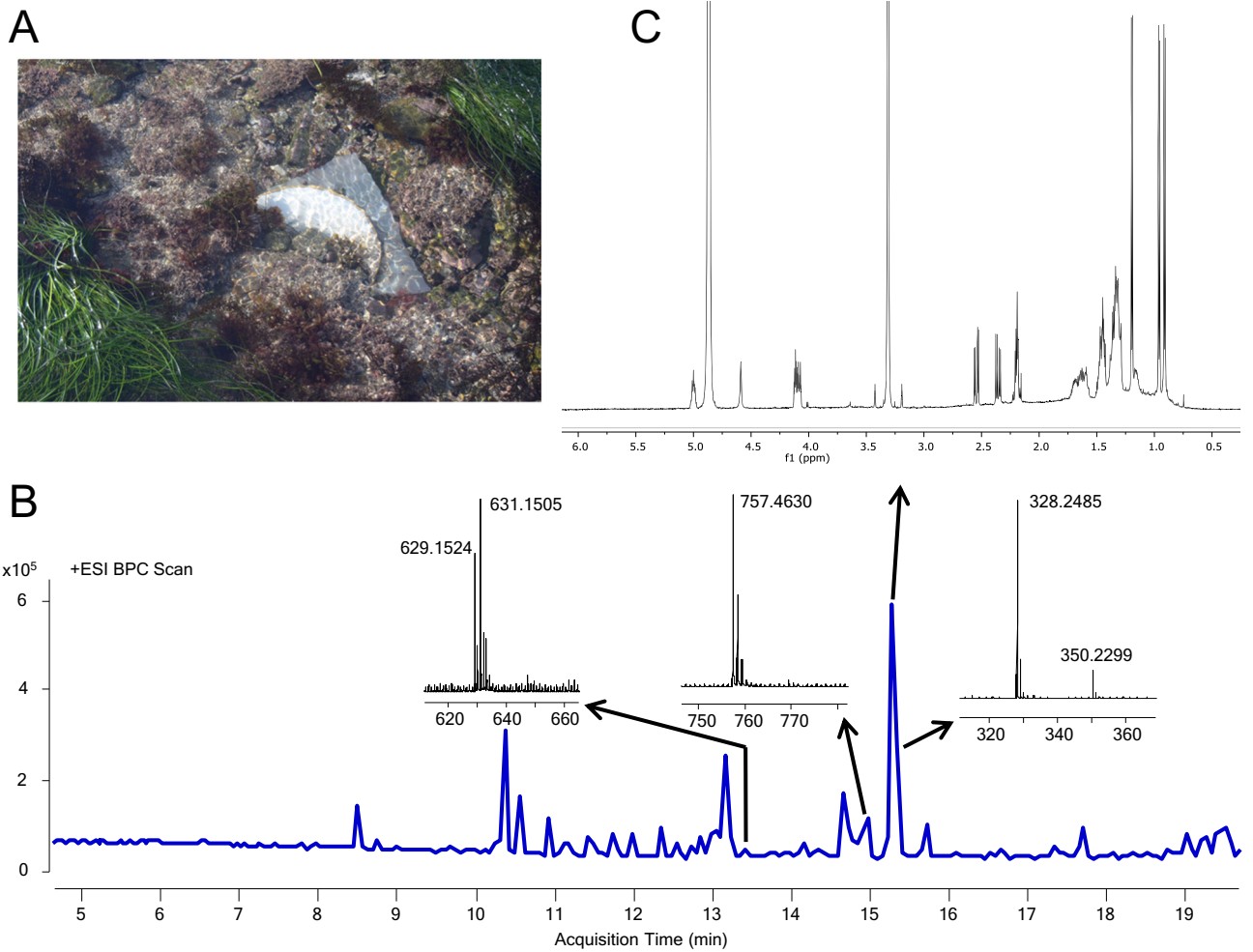

**Fig. 2 | SMIRC deployment at cabrillo state marine reserve (CSMR) and extract analysis. A** SMIRC deployed on rocky substrate. **B** Base peak chromatogram (BPC) of SMIRC extract and mass spectra of compounds targeted for isolation. ESI (electrospray ionization). **C** $^1$H NMR spectrum of *m/z* 328.2485 compound cabrillostatin (**1**, 40 µg, 600 MHz, 1.7 mm cryoprobe, CD$_3$OD). Ppm (parts per million).

**Fig. 3 | Novel compounds isolated using SMIRC.** Structures of cabrillostatin (**1**) and cabrillospirals A (**2**) and B (**3**).

additional asymmetric centers, albeit without stereocluster interconnectivity (Supplementary Figs. 7–11). The full assignment of the relative and absolute configurations remains challenging and will likely require synthetic efforts. Cabrillospiral A (**2**) eluted with two minor compounds (Supplementary Fig. 7B) that shared the same chromophore and were indistinguishable from **2** by MS/MS. After repeated rounds of HPLC purification from several extracts, we obtained one analog in sufficient amount (27 µg) to acquire $^1$H NMR, HSQC, and COSY data (Supplementary Table 4). We elucidated the structure as cabrillospiral B (**3**), which we propose as the C-5 *R* epimer of **2**.

As for many natural products, it is it's possible that **2** and **3** are post facto artifacts of subsequent chemical (or enzymatic) reactions. For example, the conjugated double bonds, exocyclic to ring A in each

of these isomeric α,β-unsaturated carbonyl compounds, may arise from a facile, non-enzymatic β-elimination of H$_2$O from precursor hemiacetals (anomeric stereocenter at C-3). Experience with furanylhemiketal natural products exposed to large excesses of MeOH solvent during extraction, suggest that O-Me ketals should appear in parallel. Detailed analysis of the LCMS data revealed no evidence of the hydrated congeners of **2** or **3** with predicated MW higher by 18 amu, nor O-Me analogs with MW incremented by 30 amu.

Notably, compounds **1, 2**, and unidentified compound *m/z* 757.4630 [M + H]$^+$ (Supplementary Information) were consistently detected in three separate deployments spanning a period of seven months, suggesting that repeat deployments can provide a path forward to address supply bottlenecks for some target compounds (Supplementary Fig. 12). Evidence that the highest compound yields were obtained from the longest deployments justifies more detailed studies of the time course of compound recovery.

## Bioactivity testing
Multiple SMIRC deployments at the CSMR site allowed us to isolate additional amounts of cabrillostatin (**1**, 26 µg) and cabrillospirals A (**2**, 75 µg) and B (**3**, 27 µg) for biological testing. Given that the amino acid statine is the pharmacophore for some protease inhibitors[39], we tested cabrillostatin (**1**) for activity against the aspartic acid protease cathepsin D. Surprisingly, **1** was not inhibitory but instead potentiated protease activity in a concentration-dependent manner from 1.5-fold at

100 nM to almost 2-fold at 10 μM (Supplementary Fig. 13A). Testing of **1** in an expanded panel of aspartic acid, serine, and cysteine proteases revealed no additional inhibitory activity (Supplementary Fig. 13B). We next tested compounds (**1**-**3**) against a panel of nine diverse cancer cell lines (Supplementary Table 5) using cell painting[40] and high-content imaging to maximize the detection of activity and infer mechanism of action[41]. Phenotypic profiles generated using a customized image analysis pipeline provided in-depth quantification of cell morphology and biomarker staining in comparison with 28 reference compounds across six diverse drug categories with distinguishable phenotypes[42] (Supplementary Fig. 14A, Supplementary Table 6). Principal component analysis (PCA) clearly separated **1** from the DMSO controls in most cell lines (Fig. 4A). Quantifying the separation showed that **1** was significantly bioactive ($p < 10^{-6}$) at 10 μM against six of the nine cell lines (Fig. 4B, Supplementary Table 7) but not against OVCAR4, HepG2, and 786-O cells indicating a degree of selectivity. The phenotypic profiles of **1** relative to the reference compounds were unique (confidence score >0.1[42], Supplementary Table 8), indicating that it possesses a distinct mechanism of action. A dose-response experiment in A549 cells validated the bioactivity of **1** at 10 μM but did not detect activity at lower concentrations (Supplementary Fig. 14B). In contrast to **1**, cabrillospirals A and B (**2**-**3**) induced no significant phenotypic changes in any of the cell lines at 10 μM.

To compliment the high-content phenotypic profiling assays, we tested compounds **1**–**3** in an induced pluripotent stem cell-derived cardiomyocyte (iPSC-CM) model[43] that can detect changes in the frequency and amplitude of cardiomyocyte beating induced by anti-arrhythmic or anti-cancer drugs (Supplementary Figs. 15A, B, Supplementary Table 9). Concordant with the phenotypic profiling results, compound **1**, but not **2** or **3**, showed a clear alteration of iPSC-CM contractions (Fig. 4C, Supplementary Fig. 15C) in the form of increased peak amplitude and a small decrease of peak frequency (Fig. 4D), both of which were validated using an independent batch of iPSC-derived cardiomyocytes (Supplementary Fig. 15D). The observed changes in the iPSC-CM $Ca^{2+}$ transients warrant further investigation given the suggested positive inotropic effect. The intriguing set of biological activities detected for cabrillostatin support the potential applications of SMIRC for natural product drug discovery, although it should be noted that special methods are required to generate defined test concentrations when dealing with small compound quantities (see Methods).

### Environmental metabolomes

We deployed SMIRC at three additional locations including a second seagrass meadow, a hypersaline lake (the Salton Sea), and a sand slope (6 m depth) off the Scripps Pier. We analyzed the extracts from all sites using GNPS classic molecular networking (CMN, Supplementary Figs. 16, 17)[44] and compared the metabolomes using FBMN[45]. A PCoA plot (Fig. 5A) revealed dramatic differences in metabolomes across sites. While more study is required to determine what drives these differences and why the CSMR extracts appeared enriched in novel compounds, these observations support deployments in diverse habitats to further explore the discovery potential of the resin capture approach. We also identified a variety of known natural products and anthropogenic contaminants, including triazine and carbamate pesticides (Supplementary Fig. 18) using GNPS molecular networking and manual database searches. Applying strict GNPS criteria (precursor ion tolerance 0.02 Da, fragment ion tolerance 0.05), we identified library matches for 325 out of 5137 (6.33%) parent ions (nodes) representing 230 unique compounds after the removal of redundant matches. The diverse compounds captured by HP-20 included fatty acids, peptides, pseudoalkaloids, tryptophan alkaloids, flavonoids, and terpenoids (Supplementary Table 10, Supplementary Fig. 17). From the Mission Bay seagrass deployment, we identified the microbial natural product apratoxin A (Supplementary Fig. 19), thus expanding the geographic distribution of this potent cyanobacterial toxin to temperate

waters[46,47]. We also identified okadaic acid and pectenotoxin-2, two polyether toxins associated with diarrhetic shellfish poisoning[48,49], as well as the macrocyclic imine toxin 13-desmethylspirolide C[50] (Supplementary Figs. 20–22). While these matches can be considered "level 2" since they are based on comparisons with literature values or MS/MS spectral libraries[51], numerous compounds detected using the DEREPLICATOR+ algorithm[52] including callipeltin B, cyanopeptolin 880, amphidinolides, kahalalide Y, and kabiramide D, could not be validated in this manner, highlighting the need for stringent manual curation of environmental metabolomes[53].

In addition to cabrillostatin (**1**) and cabrillospirals A-B (**2**-**3**), 10 compounds isolated from the CSMR extracts were considered novel based on the absence of isobaric MS matches in natural product databases. Novelty was supported by HRMS/MS data and complemented by $^{1}H$ NMR for two compounds (Fig. 5, Supplementary Table 5, Supplementary Figs. 23–32, 60, 61). Among these compounds, six display isotopic signatures indicative of chlorination (Table 1), while MS/MS fragmentation spectra indicate structural diversity that includes polyether toxins (Supplementary Figs. 23–25), alkaloids (Supplementary Figs. 26–29), peptides (Supplementary Fig. 30), and polyketides (Supplementary Figs. 31–32). Some of these compounds are associated with large molecular families (Fig. 5B), suggesting that many derivatives could also be discovered. This represents an extraordinary level of chemical novelty from a single deployment site and hints at the discovery potential of the technique.

### Environmental distributions of cabrillostatin (1) and cabrillospirals A-B (2-3)

The GNPS platform was recently expanded to include the web-enabled search engine MASST (Mass Spectrometry Search Tool)[54], the equivalent of NCBI BLAST for MS/MS datasets in the MassIVE repository. Using MASST, we parsed ~2700 untargeted, small-molecule datasets to identify the environmental distributions and potential biotic origins of cabrillostatin (**1**), cabrillospirals A-B (**2**-**3**), aplysiopsene A, and 15 other compounds (Table 1, Supplementary Tables 11, 12) detected in the SMIRC extracts. Surprisingly, **1** was identified in 11 datasets collected from 2019 to 2021 and largely associated with the study of dissolved organic matter (DOM) in nearshore[55] and deep ocean seawater samples (Supplementary Table 12). A match to a cultured dinoflagellate metabolome led us to suspect *Ostreopsis* sp. as the biogenic source. However, **1** was also detected in the medium controls, which were prepared using seawater collected from the Scripps pier[56]. Cabrillostatin clustered with several related compounds in the molecular network[55] allowing us to assign putative structures for four derivatives (cabrillostatins B, B1, C, and D) based on comparative MS/MS analyses (Fig. 5C, Supplementary Figs. 33–37). Among these, cabrillostatin B is cyclized with 11-hydroxydodecanoic acid, and cabrillostatin B1 bears an additional keto group. In cabrillostatins C and D, the statine moiety is replaced with either 4-amino-3-hydroxydeca-4-en-7,9-diynoic acid or 4-amino-3-hydroxy-5-phenylpentanoic acid (Ahppa), respectively (Supplementary Figs. 31–34). While a compound with the Ahppa unit was reported from a marine α-proteobacterium[57], the proposed diynoic amino acid is, to the best of our knowledge, unprecedented among natural products.

Determining the structure of **1** has improved our broader understanding of DOM, which has largely defied characterization[55,58], and raises questions about the ecological roles of a widespread and biologically active natural product in seawater ecosystems. Some of the compounds detected in the SMIRC extracts (Fig. 5B) are members of large and yet-to-be-characterized compound families. Annotating these families will further advance our understanding of marine DOM. In contrast to the cabrillostatins, the cabrillospirals were not detected in any of the MassIVE datasets, suggesting that this compound is highly site-specific and demonstrating the value of deploying SMIRC across diverse habitats.

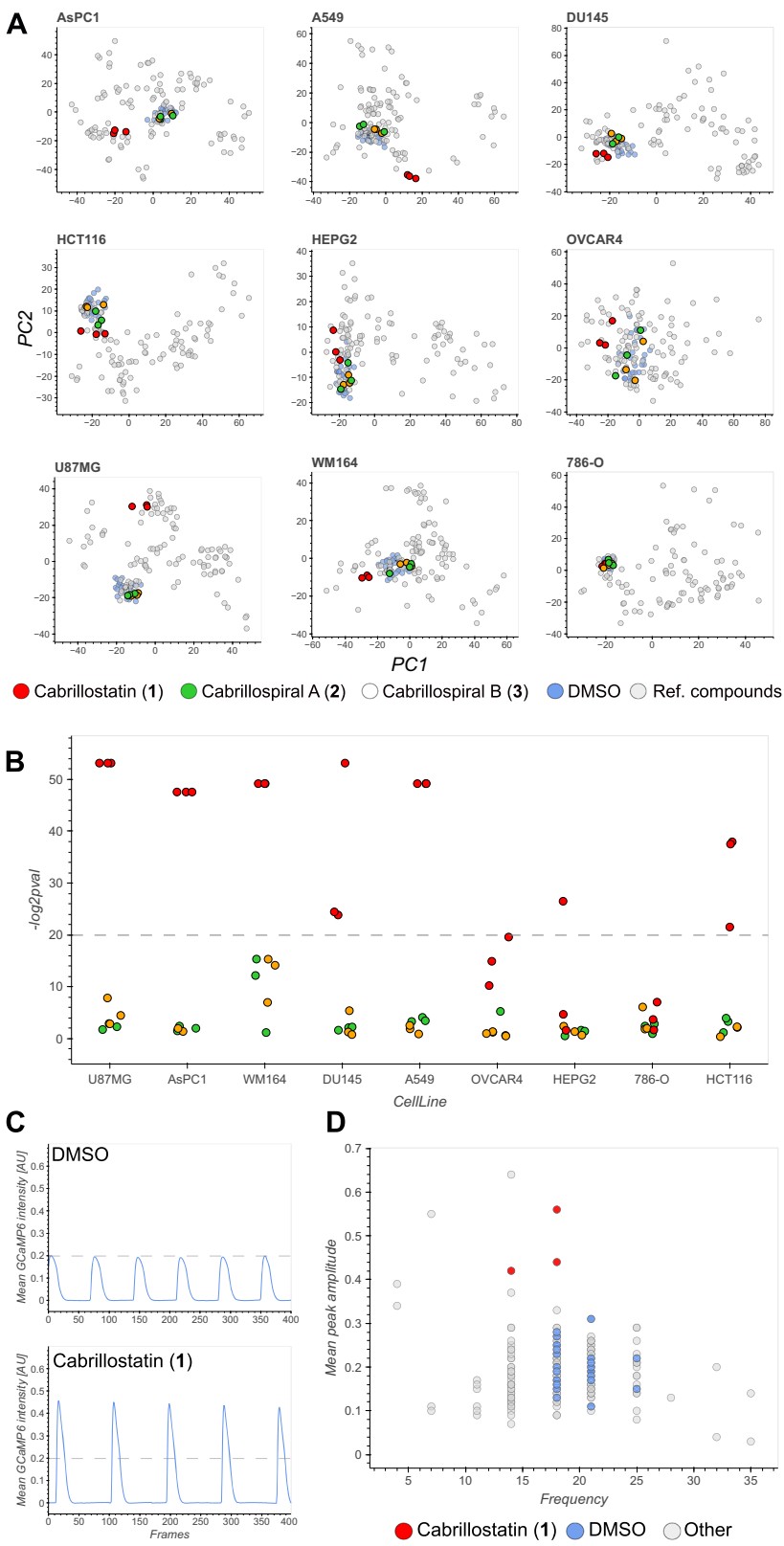

**Fig. 4 | Bioactivity of cabrillostatin (1) and cabrillospirals A (2) and B (3).**
**A** Principal component analysis of high dimensional phenotypic profiles of **1**–**3** (red, green, yellow, respectively), reference compounds (gray), and DMSO (blue) in nine diverse cell lines: AsPC 1 (human pancreas), A549 (lung), DU145 (prostate), HCT116 (colon), HEPG2 (hepatoma), OVCAR4 (ovarian), U87MG (glioblastoma), WM164 (melanoma), 786-O (renal). DMSO (dimethyl sulfoxide). **B** Bioactivity expressed as significance ($-\log^2 p$ val) across all cell lines. Responses above the

dashed line are considered significant ($p < 10^{-6}$). *P* values for compound-to-DMSO distances were calculated based on the empirical null distribution of DMSO-DMSO distances (one-sided, no adjustments for multiple comparisons; Methods).
**C** Representative Ca$^{2+}$ transient traces of DMSO (top) and **1** (bottom). **D** Scatter plot showing beating frequency and mean peak amplitude of **1** (red), DMSO (blue), and anti-arrhythmic and anti-cancer drugs (gray).

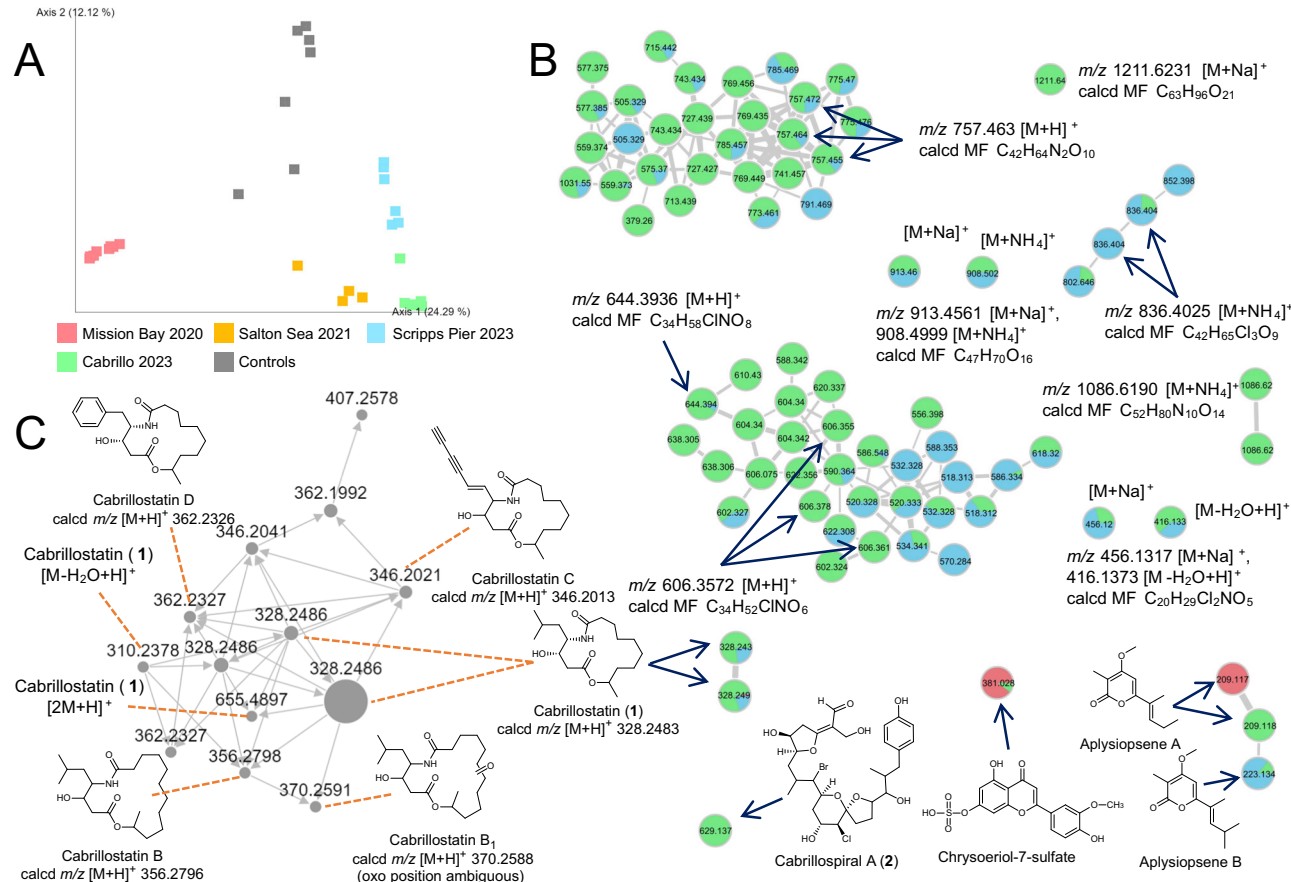

**Fig. 5 | Chemical diversity across biomes. A** PCoA plot of metabolomes derived from four locations (colors). **B** New compounds identified in the molecular network include **1-2** and others that have yet to be structurally characterized. Colors as per **A**. Numbers within nodes (circles) indicate *m/z* values. **C** Additional cabrillostatins identified in DOM based on MS/MS fragmentation spectra. MF (molecular formula), calcd (calculated).

## Metagenomic analyses

In an attempt to link compounds identified in the CSMR extracts to the producing organisms, we generated untargeted shotgun metagenomes from marine sediment and bulk seawater samples (average $76.4 \pm 12$ M paired-end reads per sample) collected at the SMIRC deployment sites. Both the sediment and seawater communities were dominated by Proteobacteria and Bacteroidetes, with higher relative abundances of Actinobacteria and Planctomycetes in the sediment samples (Supplementary Fig. 38A). In total, we recovered 1843 BGCs (N50 = 18 kbp), with many (52%) of the larger BGCs (>10 genes) predicted to be complete (i.e., not on a contig edge). The predicted products of these BGCs are largely nonribosomal peptides (28%), terpenes (25.7%), and ribosomally synthesized and post-translationally modified peptides (RiPPs, 17.7%). They could be clustered into 631 gene cluster families (GCFs, Supplementary Fig. 38B), of which only four (those encoding the alkaloid osmolyte ectoine, the peptide natural product anabaenopeptin, and the pigments flexirubin and zeaxanthin-like carotenoids) shared similarity to BGCs in the MIBiG reference database[59], emphasizing both the biosynthetic complexity and potential novelty of these environments.

Based on the structure of cabrillostatin (**1**), we expected a hybrid PKS-NRPS BGC with the starter unit adenylation (A) domain selective for leucine, a ketosynthase (KS) domain that could account for the $C_2$ elongation of leucine to yield the statine moiety, coupled with highly reducing PKS modules to account for the saturated fatty acyl component. However, none of the 43 PKS-NRPS hybrid BGCs recovered from the two sample types encoded the predicted enzymes. Given that a low abundance organism could account for production, we compared KS domains from the unassembled reads with those associated with statine biosynthesis in the didemnin B and burkholdac A BGCs[36,60], however none clustered in a KS phylogeny. Interestingly, the two reference KSs shared <60% amino-acid identity with a database of metagenome-extracted KSs, supporting the rarity of statine incorporation into natural products.

For cabrillospirals (**2-3**), potential biosynthetic hooks included a modular type I PKS (T1PKS), halogenases, a *p*-hydroxybenzoic acid (*p*-HBA) starter unit, and methyltransferases to account for the C-2 hydroxymethyl and C-8 methyl groups. Among the 68 T1PKS BGCs identified, we detected two candidates (Supplementary Fig. 38C) in a high-quality MAG (4.96 Mbp; 98.2% complete, 3.6% contamination) in the phylum Planctomycetes, which was recently recognized as a rich and poorly explored source of specialized metabolites[61]. While lacking key biosynthetic genes, the planctomycete MAG was nonetheless the best candidate identified. Unfortunately, long-read Oxford Nanopore sequencing failed to cover this BGC and efforts to culture Planctomycetes were unsuccessful[35]. While genetic linkages between compounds and producers have been established in the context of symbioses[35,62], the results presented here reinforce the challenges associated with making these connections in complex microbial communities.

## Discussion

Many of today's most beneficial medicines are derived from natural products. The discovery process has invariably started with the extraction of collected or cultivated organisms followed by chemical or bioassay-guided compound isolation. While effective for decades, it has become increasingly difficult to find new carbon skeletons using these traditional approaches. The realization that microbial genome sequences contain significant, unrealized biosynthetic potential

**Table 1 | Additional compounds isolated from the CSMR site**

| z, ionization | Molecular formula (MF) | MF matches on DNP | Predicted compound class (CANOPUS annotation) | Highest substructure Tanimoto similarity |
|---|---|---|---|---|
| 757.4630, [M + H]$^+$ | $C_{42}H_{64}N_2O_{10}$ | none | Nitrogen containing polyether (cerveratrum alkaloid, steroidal alkaloids) | 52.31% |
| 456.1317, [M+Na]$^+$ 416.1373, [M-H$_2$O + H]$^+$ | $C_{20}H_{29}Cl_2NO_5$ | none | Chlorinated alkaloids (aryl chlorides, polyketides) | 31.48% |
| 1211.6324, [M+Na]$^+$ 1206.6796, [M + NH$_4$]$^+$ | $C_{63}H_{96}O_{21}$ | Spongistatin 2, steroidal glycosides | Polyether polyketide (triterpenoids) | NA |
| 1086.6190, [M + NH$_4$]$^+$ 1069.5915, [M + H]$^+$ | $C_{52}H_{80}N_{10}O_{14}$ | none | Peptide (cyclic depsipeptides) | 56.90% |
| 836.4044 [M + NH$_4$]$^+$ | $C_{42}H_{65}Cl_3O_9$ | none | Halogenated polyketide, likely related to **2** | NA |
| 488.257, [M + H]$^+$ | $C_{23}H_{34}ClNO_8$ | none | Chlorinated alkaloids (alpha-acyloxy ketones, open-chain polyketides, vinylogous esters) | 43.25% |
| 606.3572, [M + H]$^+$ | $C_{34}H_{52}ClNO_6$ | none | Chlorinated alkaloids (N-acyl amines, open-chain polyketide) | 39.04% |
| 644.3936, [M + H]$^+$ | $C_{34}H_{58}ClNO_8$ | none | Chlorinated alkaloids (N-acyl amines, open-chain polyketide) | 39.32% |
| 730.2286, [M + Na]$^+$ | $C_{33}H_{48}Cl_3NO_9$ | none | Chlorinated alkaloids (macrolactams, ansa macrolides) | NA |
| 913.4561, [M+Na]$^+$ 908.4999, [M + NH$_4$] | $C_{47}H_{70}O_{16}$ | PF 1025, desfontainic acid, steroidal glycosides | Polyether polyketide | 70% |

*DNP* dictionary of natural products. Bold number refers to cabrillo spiral A.

launched the field of genome mining and efforts to activate silent BGCs through co-cultivation, elicitor addition, or genetic manipulation[18]. Yet despite global efforts, it has been estimated that heterologous expression has only yielded 12 new compound families/yr over the last five years, of which half were ribosomally encoded peptides[63]. Given these incremental advances, it remains clear that current approaches to natural product discovery have failed to capture the vast extent of chemical diversity encoded by Earth's microbiomes.

Here we describe a compound-first approach to natural product discovery in which adsorbent resins capture compounds directly from the environments in which they are produced. Using this technique, which we call SMIRC, natural products were captured and purified from a marine deployment site in sufficient quantities to elucidate the structures of cabrillostatin (**1**) and the halogenated polyketides cabrillospirals A and B (**2**-**3**), which represent two new carbon skeletons. The latter possesses a functional group (vinylogous formate ester) that is unprecedented in both the natural product and synthetic literature. The Tanimoto similarity coefficient calculated for cabrillospiral A (0.47), which describes the similarity of this compound to other structures, demonstrates its uniqueness among natural products. The isolation and partial characterization of many other new compounds from the CSMR location hints at the potential of this technique to access new chemical space, a high priority for natural product research. Why this site appeared particularly rich in novel compounds remains unknown, but it's intriguing to suggest that increased biodiversity and reduced anthropogenic disturbance within a marine protected area may have played a role.

The biological activities associated with cabrillostatin (**1**), including phenotypic changes in cancer cells and effects on cardiomyocyte beat frequency and amplitude, demonstrate the applications of this approach to natural product drug discovery. While the reference compounds tested were sufficiently distinct from cabrillostatin to suggest a unique mode of action, we note that drugs with similar effects (e.g., entinostat) are HDAC inhibitors. The similarity of cabrillostatin with HDAC inhibitors is also seen in the effects on cardiomyocytes, where comparable alterations of beating frequency and amplitude were observed. While it remains to be determined if cabrillostatin is an HDAC inhibitor, it is interesting to note that this biologically active compound is a widespread component of seawater DOM that can now be studied in an ecological context and used, as we have shown, to facilitate the identification of related compounds.

While our efforts to link these compounds to their biological origins remain ongoing, establishing these types of linkages in complex systems such as the ones sampled here remains a major challenge.

While SMIRC will benefit from further refinements, we show that compounds can be recovered in sufficient quantities for NMR-based structure elucidation and bioactivity testing. Re-supply challenges remain the major bottleneck and will need to be addressed on a case-by-case basis. The recovery of some compounds from the same location over time indicates that re-deployments can be used to obtain additional material, as we showed for cabrillostatin and the cabrillospirals at the CSMR site. Nonetheless, temporal variability in compound recovery remains an important consideration when selecting the initial quantity of resin to deploy. Incorporating resin in agar to support in situ microbial growth led to the isolation of a relatively large amount of aplysiopsene A and may increase the likelihood that producing strains can be obtained in culture, although more work is needed to explore this concept. Synthesis represents a tractable approach to supply compounds such as cabrillostatin, while more complex structures such as the cabrillospirals will require a larger investment from the synthetic community or beyond. Regardless of how compound re-supply is addressed, the discovery of new natural product chemotypes remains a top priority for natural products research and a driving force in drug discovery.

In summary, SMIRC provides a unique approach to access chemical space that has remained beyond reach using current methods. It is agnostic of the producing organism and can be used to isolate novel natural products directly from Nature in sufficient quantities for NMR-based structure elucidation and biological testing. This compound-first approach inverts the traditional paradigm of microbial natural product discovery, thus circumventing many of the major bottlenecks associated with cultivation and genome mining, although it has its own unique set of limitations that need to be addressed through further method development. By incorporating up-front bioassays, bioactive compounds can be directly targeted without the need to build large culture collections. SMIRC is relatively simple to assemble and deploy, requiring the type of analytical instrumentation readily available at most research-intensive universities. Moving forward, this approach can be expanded to other environments, provide opportunities to explore the ecology of chemically mediated interactions, and generate unique chemical libraries for drug discovery and other commercial efforts that benefit from natural product diversity.

## Methods

### Resin processing and extraction

HP-20 resin was soaked in MeOH, gently shaken at 40 rpm for 1 h, then filtered (Büchner funnel-filter membrane, 200 μm) under reduced pressure and washed with MeOH (x3) then $H_2O$ (x3). HPLC grade MeOH and MilliQ grade $H_2O$ were used throughout, and all glassware was acid washed and rinsed with MeOH before use. Bulk-washed resin could be stored in MilliQ water at room temperature for several weeks before deployment, at which time 100 g was packaged between two layers of Nitex™ nylon mesh (120 μm, Genesee Scientific, El Cajon, CA) and framed within a 30 cm wooden embroidery hoop (Fig. 1A). Packaged resin was secured in the field using aluminum tent stakes, rocks, or attached to ropes ~30 cm above the seafloor using SCUBA. Upon recovery, individual resin units were washed with deionized water and the resin removed from the nylon mesh and transferred into a glass fritted Büchner funnel (200 μm pore size), where it was rinsed with $H_2O$ and eluted three times with 200 mL MeOH under vaccuum. The solvent was removed on a rotary evaporator to yield a salt-free crude extract. Agar-resin matrix was prepared by adding 250 g of washed HP-20 resin to 33 g Instant Ocean salt mixture and 12 g agar in 1 L of deionized $H_2O$, which was autoclaved at 121 °C for 20 mins. After cooling to 50 °C, the liquid agar-resin was swirled and distributed into ten 15 cm Petri dishes. Two solidified agar/resin plates were aseptically removed from the dishes and enclosed between two layers of Nitex™ nylon mesh as described above. The Nitex™ mesh and embroidery hoops were treated with 70% isopropanol and allowed to dry before packaging. Packaged containers were wrapped with 70% isopropanol-treated aluminum foil prior to deployment. Upon recovery, agar-resin from each container was extracted three times with 150 mL of dichloromethane-MeOH (1:1) and the solvents removed on a rotary evaporator.

### LCMS analyses

LC-HRMS was performed on an Agilent 6530 Accurate-Mass QToF with ESI-source coupled with an Agilent 1260 Infinity HPLC equipped with a degasser, binary pump, autosampler, DAD detector, and a 150 × 4.6 mm Kinetex $C_{18}$ 5 μm column (Phenomenex, Torrance, CA) calibrated using the Agilent Reference Calibration Mix. A total number of 108 samples including crude extracts ($n = 51$), SPE fractions ($n = 34$) and controls ($n = 23$) were analyzed. Controls included non-deployed HP-20 resin and Nitex mesh MeOH extracts and the solvents used for extraction and treatment of the SMIRC containers (MeOH, DCM, isopropanol). The samples were dissolved, when applicable, in MeOH (5 μL, 1 mg/mL) and eluted (1 mL/min) using an isocratic mobile phase (20:80 acetonitrile (ACN):$H_2O$ and 0.1 % formic acid (FA)) for 2 min followed by a gradient elution to 95:5 ACN:$H_2O$ over 18 min, isocratic for 2 min, then increasing to 100% ACN over 1 min, and finishing with 100% ACN for 2 min. For feature-based molecular networking, a 100 × 3 mm Kinetex $C_{18}$ 2.6 μm particle size column (Phenomenex, Torrance, CA) was used. Crude extracts ($n = 24$) and controls ($n = 7$) were eluted using an isocratic mobile phase 10:90 ACN:$H_2O$ and 0.1 % FA for 2 min followed by a gradient elution to 98:2 ACN:$H_2O$ over 20 min, then increasing to 100% ACN over 3 min and finishing with 100% ACN for 1 min. The flow was set to 0.5 mL/min.

MS data were acquired over the range 135–1700 $m/z$ in positive mode using the following parameters: MS scan rate, 2/s; MS/MS scan rate, 3/s; gradient collision energy (slope 2.6, offset 15 eV); source gas temperature, 300 °C; gas flow, 11 L/min. All solvents were LCMS grade. The data was manually analyzed using the Masshunter Qualitative Analysis Software B.05.00.

### Molecular networking and feature detection

CMN and automated library searches were performed using the online workflow (https://ccms-ucsd.github.io/GNPSDocumentation/) on the GNPS website (http://gnps.ucsd.edu)[64] and visualized using Cytoscape

3.8.2[65] and the GNPS network visualizer. For CMN, the data was filtered by removing all MS/MS fragment ions within ±17 Da of the precursor $m/z$. MS/MS spectra were filtered by choosing the top 6 fragment ions in the ±50 Da window throughout the spectrum. The following settings were used: precursor ion mass tolerance (PIMT) 0.02 Da, MS/MS fragment mass tolerance (FMT) 0.05 Da, and cosine score cutoff 0.7 with more than three matched fragments. The maximum size of a molecular family was set to 100. The library spectra were similary filtered with matches requiring a cosine score >0.7 and at least four matched fragment peaks. Spectra of interest were queried in MASST via the GNPS platform[54] and publicly available metabolomic datasets were searched with default parameters (https://masst.ucsd.edu/). For the FBMN, LCMS raw data was pre-processed with MZmine 2.53[66] using MS1 feature detection: retention time (rt) 2–25 min, MS signal intensity 1E$^3$, centroid mass detector; MS2 feature detection: rt 2–25 min, MS signal intensity 1E$^2$, centroid mass detector. MS1 features were transformed to chromatograms with the minimum number of scans 2, group intensity threshold 1E$^3$, minimum highest intensity 2E$^3$, $m/z$ tolerance 0.05 Da or 20 ppm. Chromatograms were deconvoluted using the baseline cutoff algorithm (min peak height 4E$^3$, peak duration 0.02–1.0 min, baseline level 1E$^3$), $m/z$ center calculation was set to MEDIAN, $m/z$ range for MS2 scan pairing was set to 0.02 Da and rt range to 0.1 min. The chromatograms were deisotoped with $m/z$ tolerance of 0.02 Da or 10 ppm, absolute rt tolerance of 0.1 min, maximum charge 2, and the representative isotope was set to lowest $m/z$. Aligned feature lists were built with the join aligner, $m/z$ tolerance of 0.02 Da, rt tolerance of 0.2 min, and weight for $m/z$ and rt set to 75 and 25, respectively. The aligned feature list was exported to FBMN. The network was created with a PIMT 0.1 Da, FMT 0.1 Da, cosine cutoff 0.7, and 4 matched peaks, the maximum size of a molecular family 200, while other settings were the same as for CMN above. Both FBMN and CMN jobs including detailed settings description can be accessed online under https://gnps.ucsd.edu/ProteoSAFe/status.jsp?task=26bfbaebdd7245c182579769ee258822 (CMN) and (https://gnps.ucsd.edu/ProteoSAFe/status.jsp?task=8130a98b878c4c6fa55c8ef3e412c648) (FBMN).

### Molecular formula annotation

The molecular formulas for the compounds of interest were determined using the online tool ChemCalc[67]. MS/MS spectra of targeted compounds were exported as .mgf files from Masshunter Qualitative Analysis Software B.05.00 and manual curation applied to remove noise and $^{13}C$ and $^{37}Cl$ isotopic mass peaks. The .mgf files were then subjected to the SIRIUS 5.8.3 analysis platform, which includes CSI:FingerID substructure prediction[68], and CANOPUS[69], followed by further manual analysis.

### Antibiotic screening

Crude extracts were screened for antibiotic activity using an HPLC microfractionation assay. In brief, 1 mg (i.e., 33 μL at 30 mg/mL) of crude extract was injected into an analytical HPLC (reversed-phase 150 × 4.6 mm Kinetex $C_{18}$ 5 μm column, flowrate 0.8 mL/min) using a gradient from 10–100% ACN (with 0.1 % FA) over 16 min then isocratic for 4 min. Fractions (200 μL) were collected every 15 s starting at 1.8 min (system dead time) into flat bottom 96-well micro-titer plates using a Gilson FC204 fraction collector, yielding 80 fractions per extract. The fractions were dried under vacuum (4 h), and the content of each well dissolved in 10 μL DMSO and 190 μL of an overnight culture of outer membrane defective *E. coli* LptD4213[30,70]. Positive (10 μM chloramphenicol) and negative (10 μM streptomycin and solvent/media) controls were tested for each plate. The plates were incubated for 16 h with shaking (120 rpm, 30 °C) and turbidity (OD$_{650}$) measured before and after incubation on a Molecular Devices Emax Precision Plate Reader (Molecular Devices, San Jose, CA). Growths inhibition was calculated as the reduction in OD relative to solvent/media controls.

## Compound isolation

For chryseriol, crude extracts were separated into five fractions using 2 g $C_{18}$ reversed-phase silica gel 90 Å pore size (Sigma-Aldrich, St. Louis, MO) and a 20 mL MeOH:$H_2O$ step gradient (25:75, 50:50, 75:25, 90:10, and 100:0 MeOH:$H_2O$). HPLC was performed using Agilent's 1100 G1312A binary pump, 1100 G1315A DAD UV/Vis detector, 1100 G1313A autosampler, and 1100 G1322A degasser (Agilent Technologies, Santa Clara, CA) coupled to a Shimadzu (Kyoto, Japan) low-temperature ELSD through a Hewlett Packard 35900E (Spring, TX) interface. Antibiotically active fractions were further separated by HPLC (10 × 250 mm $C_8$ column, isocratic MeOH:$H_2O$ (69:31) with 0.1% FA) and 2.5 mL/min flow. For aplysiopsene A, crude agar-resin extract was fractionated following the same procedure and fraction 3 (75:25 MeOH:$H_2O$) further separated by HPLC (4.6 × 150 mm $C_{18}$ column, isocratic ACN:$H_2O$ (50:50) with 0.05% FA, 1 mL/min flow). For cabrillostatin (**1**) and cabrillospirals A-B (**2-3**), crude extracts from four resin containers were separately fractionated as described above and respective fractions 3 (75:25 MeOH:$H_2O$) were then submitted to HPLC (4.6×150 mm $C_{18}$ column, isocratic ACN:$H_2O$ (35:65) with 0.05% FA, 1 mL/min flow) for isolation of **1-3**.

## NMR and ECD spectroscopy

NMR spectra (1D and 2D) were measured at 23 °C on a JEOL ECZ spectrometer (500 MHz) equipped with a 5 mm $^1H$ room temperature probe (JEOL, Akishima, Tokyo, Japan) or on a Bruker Avance III (600 MHz) NMR spectrometer with a 1.7 mm $^1H[^{13}C/^{15}N]$ microcryoprobe (Billerica, MA). Quantitation of sub-milligram samples was achieved using the 'quantitative solvent $^{13}C$-satellite' method, which is estimated to have 5–10% accuracy, and cholesterol as the external standard[71]. NMR spectra were referenced to the solvent signals ($CHD_2OD$, $\delta_H$ 3.31, $CD_3OD$, $\delta_C$ 49.00 ppm; $CHD_2CN$, $\delta_H$ 1.94, $CD_3CN$, $\delta_C$ 1.74 ppm). The NMR spectra were processed using MestReNova (Mnova 12.0, Mestrelab Research). ECD spectra were measured on a JASCO J-810 spectropolarimeter (JASCO, Tokyo, Japan) at 23 °C in $CH_3OH$ and 5 mm pathlength quartz cells. Minimal energy conformers of model compounds were constructed with Spartan '20 or '24 (Wavefunction, Inc., Irvine, CA, USA), and the ECD spectra were calculated with Gaussian 09 Revision-C.01-SMP (Gaussian, Inc., Walligford, CT, USA).

Detailed structural elucidation of compounds **1-3** is provided in the Supplementary Notes.

## Quantification of flavonoids in seawater

One liter of seawater was collected from the seagrass meadow, passed through a Waters Oasis HLB SPE column (Waters Corporation, Milford, MA), eluted with 3 mL MeOH, and analyzed by HPLC (UV detection, 360 nm) using chromatography conditions described above for antibiotic screening (microfractionation). A methanolic solution of purified chryseriol (1 mg/mL) was used as a standard. AUC values were used to calculate flavonoid concentrations.

## Cell lines

Lung (A549, #CCL-185), prostatic (DU145, #HTB-81), renal (786-O, #CRL-1932), and pancreatic (AsPC1, #CRL-1682) carcinoma cells were purchased from ATCC (Manassas, VA). Hepatocellular (HEPG2, #CCLZR209), glioblastoma (U87MG, #CCLZR406), and colorectal (HCT116, #CCLZR253) carcinoma cells were purchased from the UCSF cell line core facility. Ovarian (OVCAR4, #SCC258) carcinoma cells were purchased from Sigma-Aldrich (St. Louis, MO). WM164 was a gift from Dr. Helmut Schaider. A549, OVCAR4, 786-O, WM164, and AsPC1 were maintained in RPMI-1640 (Invitrogen, Waltham, MA). DU145, HEPG2, U87MG, and HCT116 were maintained in DMEM with 4.5 g/dL glucose (Invitrogen, Waltham, MA). Both media were supplemented with 10% fetal bovine serum (Gemini Bio, West Sacramento, CA) and 1% penicillin/streptomycin (Thermo Fisher Scientific, Waltham, MA). Cell lines

were maintained in culture in a humidified 37 °C incubator with 5% $CO_2$ for a maximum of 6 weeks. Frozen cell lines were stored in liquid nitrogen with 10% DMSO. All cell lines were routinely tested by PCR for mycoplasma[72,73]. No known misidentified cell lines were used, and cell lines were authenticated in 2024 using Short Tandem Repeat profiling.

## Cell painting

Reference compounds were purchased from Selleck (Houston, TX) and/or MedChemExpress (Monmouth Junction, NJ) and tested at 2 doses (1:1000 and 1:10,000 dilutions) in duplicate. Natural products were screened at 10 μM in triplicate. Cells were seeded into 384-well PhenoPlates (Perkin Elmer, Waltham, MA) at an empirically determined density in 75 mL of the respective media per well and treated with compounds for 48 h. Compounds were added directly to the well using the ECHO 650 liquid handling system (Beckman Coulter, Brea, CA) integrated into a Perkin Elmer EXPLORER G3 WORKSTATION. Transfers and plate handling were established using Perkin Elmer's plate::works™ (v6.2) software. Staining with cell painting dyes were adjusted from prior studies[40]. Briefly, 10 μL of an 8× Mitotracker deep red (Invitrogen, Waltham, MA) master mix (in DMEM at a concentration of 5 mM) was added to each well, incubated for 30 min, then fixed by adding 30 μL 16% paraformaldehyde (final concentration 4%) for 30 min at room temperature (RT). Plates were then washed with 1× HBSS (Invitrogen, Waltham, MA), permeabilized with 1× HBSS + 0.5% (vol/vol) Triton X-100 solution (30 min), and washed twice with 1× HBSS. Thirty ml of a staining master mix in 1× HBSS + 1% (wt/vol) BSA solution (Supplementary Table 1) was added and incubated in the dark at RT for 30 min. All dyes were purchased from Life Technologies. Finally, cells were washed three times with 1× HBSS, covered with aluminum foil seals, and light-protected until imaging. All pipetting steps were automated using the Perkin Elmer EXPLORER G3 WORKSTATION and performed by MultiFloFX and 405 TS washer (BioTek, Agilent Technologies).

## High-content imaging and data analysis

Imaging was performed on the Operetta CLS confocal spinning-disk high-content analysis system (Perkin Elmer, Waltham, MA) using a ×20 water-immersion objective. Each well was imaged at 5 fields of view, in 5 channels (filters according to wavelength information in Supplementary Table 13) in 3 z-planes. Cell segmentation and single-cell feature extraction were performed using the Harmony™ software (v4.9, Perkin Elmer, Waltham, MA) from maximum intensity projections of each field of view. In total, -1100 features were calculated for each cell, including intensity, morphology, and texture features. Phenotypic profiles were calculated using the Kolmogorov–Smirnov (KS) statistic and bioactivity defined as a cellular responses distinct from DMSO controls[42]. The Mahalanobis metric was used to calculate distances between DMSO replicates and between DMSO and test compounds. A Gaussian kernel estimation was then applied to the DMSO-DMSO distances to establish a null distribution, which was compared to the DMSO-compound distance. Compound distances ≥2X the 99th percentile of the null distribution ($p \leq 10^{-6}$) were defined as bioactive. Reference compound classification and novel compound class prediction were calculated per well based on phenotypic profiles[42]. All computations were performed using Python (v3.9).

## iPSC-derived cardiomyocyte differentiation

Induced pluripotent stem cell (iPSC) derived cardiomyocytes were generated from human IPSCs (WTC background genetically modified to inducibly express dCas9-KRAB fusion protein and constitutively express GCaMP6, purchased from the Gladstone Institute Stem Cell Core facility) using previously published differentiation, expansion, and maturation protocols with modifications[43,74,75]. Briefly, human iPSCs were dissociated into a single-cell suspension using TrypLE™ Express (Thermo Fisher Scientific, Waltham, MA) and seeded into

Geltrex™ LDEV-Free Reduced Growth Factor Basement Membrane Matrix-coated 12 well-plates (Thermo Fisher Scientific, Waltham, MA) at 75k cells per well in E8 media supplemented with 10 μM Y-27632. Cells were fed daily with E8 media and differentiation was initiated once the iPSCs reached 80–90% confluence (~48 h) by addition of RPMI-1640 supplemented with B27 without insulin and 7.5 μM CHIR99021. After 48 h, the media was replaced with RPMI-1640 supplemented with B27 without insulin and 7.5 μM IWP2. After an additional 48 h, the media was replaced with RPMI-1640 supplemented with B27 without insulin. Two days later, the media was changed to RPMI-1640 supplemented with B27 containing insulin and 1% P/S that was replenished every 2-3 days. Spontaneous beating was generally observed 8–10 days after the addition of CHIR99021. iPSC-derived cardiomyocytes (iCMs) from wells that showed spontaneous beating in >80% of the surface area were exposed to lactate enrichment medium. After 6 days of enrichment, iCMs were seeded into Geltrex coated 72 cm² flasks and cultured in RPMI-1640 with B27, insulin, and 2 μM CHIR99021 until reaching confluence. iCMs were then replated into Geltrex coated 384-well PhenoPlates (Perkin Elmer, Waltham, MA) and switched to maturation medium for 10 days prior to use.

### iPSC-derived cardiomyocyte Ca²⁺ transient imaging and analysis

Anti-arrhythmic and anti-cancer drugs were purchased from Selleck (Houston, TX) or MedChemExpress (Monmouth Junction, NJ) and tested at 2 doses (1:1000 and 1:10,000 dilutions) in duplicate. Natural products were screened at 10 μM in triplicate. After 24 h or 48 h of drug treatment (primary and validation experiments, respectively), iPSC-derived cardiomyocyte Ca²⁺ transients were monitored using an EVOS M7000 Imaging System (Thermo Fisher Scientific, Waltham, MA) with atmospheric control set at 37 °C, 80% humidity, 5% CO2, for 20 s at a frame rate of 30hz in the GFP channel (Excitation max 488 nm) using a ×20 objective (one FoV). Mean fluorescence intensities for each field of view were extracted using the scikit-image (v1.2.2) module. The signal was baseline-corrected using the Python package BaselineRemoval (v0.1.3), and the mean signal intensity of frames 200–600 (~13 sec) was plotted for each compound.

### Protease inhibition studies

SARS-CoV-2 Mpro was expressed as outlined previously[76]. SARS-CoV-2 PLpro was purchased from Acro Biosystems (PAE-C518), and human TMPRSS2 was purchased from Cusabio Technology (CSB-YP023924HU). Other recombinant proteases were purchased from R&D Systems, which included SARS-CoV-1 3CL/Mpro (E-718), MERS-CoV 3CL/Mpro (E-719), human cathepsin B (953-CY), human cathepsin L (952-CY), human cathepsin D (1014-AS), and human cathepsin E (1294-AS). The final enzyme concentration in each assay was 50 nM of SARS-CoV-1, SARS-CoV-2 Mpro, MERS-CoV Mpro, and SARS-CoV-2 PLpro; 0.5 nM of human cathepsin L and B, 1 nM human cathepsin D and E, and 3 nM TMPRSS2[77]. Fluorogenic peptide substrates were purchased from a variety of vendors and used at the following concentrations: 10 μM MCA-Ala-Val-Leu-Gln-Ser-Gly-Phe-Arg-Lys(DNP)-Lys-NH₂ (a gift from Charles Craik, University of California San Francisco)[78], 50 μM Mu-His-Ser-Ser-Lys-Leu-Gln-AMC (Sigma SCP0224), 50 μM z-Arg-Leu-Arg-Gly-Gly-AMC (Bachem I1690), 5 μM Mca-Gly-Lys-Pro-Ile-Leu-Phe-Phe-Arg-Leu-Lys(DNP)-DArg-NH2 (CPC Scientific, SUBS-017A), 85.5 μM Boc-Gln-Ala-Arg-AMC (Biosynth, MQR-3135-v), 5 μM z-Phe-Arg-AMC (R&D Systems, ES009) for cathepsin L, and 100 μM of the same substrate for cathepsin B. Assay buffers for protease assays were as follows: Assay Buffer 1 for SARS-CoV-1 Mpro and MERS-CoV Mpro: 50 mM HEPES pH 7.5, 150 mM NaCl, 1 mM EDTA, 1 mM DTT, 0.01% Tween-20; Assay Buffer 2 for SARS-CoV-2 Mpro: 50 mM Tris-HCl, pH 7.5, 150 mM NaCl, 1 mM EDTA, 0.01% Tween-20; Assay Buffer 3 for SARS-CoV-2 PLpro: 50 mM HEPES pH 6.5, 150 mM NaCl, 0.01% Tween-20, 0.1 mM DTT; Assay Buffer 4 for Cathepsin B:

50 mM Na-Acetate pH 5.5, 5 mM DTT, 1 mM EDTA, 0.01 % BSA; Assay Buffer 5 for Cathepsin L: 50 mM Na-Acetate pH 5.5, 5 mM DTT, 1 mM EDTA, 0.01 % BSA, 100 mM NaCl; Assay Buffer 6 for Cathepsin D and E: 50 mM Citrate phosphate buffer, and Assay Buffer 7 for TMPRSS2: 25 mM Tris-HCl, pH 8.0, 150 mM NaCl, 5 mM CaCl₂, 0.01% Triton X-100. Cabrillostatin (1) was diluted into the appropriate assay buffer to a concentration of 40 μM and then pre-incubated with each protease for 15 min at 25 °C. Separately, the fluorogenic substrates were diluted in the appropriate assay buffers at twice the target concentration and then added to the enzyme/inhibitor mixture. All assays were performed at 25 °C in triplicate wells, and DMSO was used as vehicle control. The final test concentration of cabrillostatin was 10 μM. Control inhibition assays were conducted with 10 μM GC373 for all 3CL/Mpro enzymes (a gift from John Vederas, University of Alberta), 10 μM of compound 159, an in-house discovered PLpro inhibitor[78], 10 μM E-64 for cathepsin L and B (Sigma E3132), 10 μM Pepstatin for cathepsin D and E (Sigma P5318), and 1 μM Camostat (a gift from James Janetka, Washington University St. Louis). The final volume of each reaction was 30 μL in a 384-well plate, and fluorescence was measured at 360/460 nm (ex/em) for peptide-AMC substrates or 320/400 nm (ex/em) for internally quenched MCA-peptide-DNP substrates in a Biotek Synergy HTX fluorescence plate reader. The reaction velocity was calculated as relative fluorescent units per second. The activity was normalized to wells lacking inhibitor but containing the vehicle (DMSO) in assay buffer.

### Metagenomic analyses

Sediment (February and March 2021) and bulk seawater samples (August 2021) were collected in parallel with the SMIRC deployments. Sediment samples (~80 g) were collected in sterile WhirlPak™ bags and transported on ice to Scripps Institution of Oceanography for storage. Bulk seawater samples (2 L each) were immediately filtered (0.2 μm Sterivex). All samples were stored (−20 °C) until further processing. DNA was extracted using a phenol-chloroform protocol[79] and assessed using a Nanodrop for quality and concentration estimates. When needed, low-concentration extracts from the same sediment/water samples were pooled prior to sequencing. Libraries for all samples were prepared using the Nextera XT DNA library preparation kit (Illumina, San Diego, CA) for sequencing on an Illumina NovaSeq 6000 system with 150-bp paired-end reads.

Raw reads were quality trimmed, and adapters were removed using the BBMap package. For taxonomic and biosynthetic analyses, samples were processed using a read-based approach to limit assembly biases[80]. Briefly, single-copy phylogenetic marker genes were extracted with hidden Markov models (HMMs)[81] from each sample and classified against a reference genomic database using phylogenetic inferences with pplacer (v1.1.alpha17)[82]. Translated reads were searched against the NaPDoS2[83] database for ketosynthase domains. Filtered reads were normalized with BBNorm (target = 40, mindepth = 5) and assembled with the IDBA-UD assembler using the pre-correction flag (mink=30, maxk=200, step=10)[84]. Assembled contigs (>2000 bp) were searched for BGCs with antiSMASH (v5.1.2; both fungal and bacterial versions)[85]. BGCs were clustered into gene cluster families (GCFs) based on relatedness using BiG-SCAPE (v1.1.2)[86]. Contigs were binned into metagenome-assembled genomes (MAGs) by calculating coverage profiles with bowtie2[87] followed by an unsupervised binning approach with Metabat2 across individual samples[88]. All MAGs were checked for quality (i.e., completeness and contamination) with checkM[89], dereplicated with dRep[90], and assigned taxonomy with GTDB-Tk[91].

### Reporting summary

Further information on research design is available in the Nature Portfolio Reporting Summary linked to this article.

## Data availability

The LCMS data generated in this study can be accessed on the Mass spectrometry Interactive Virtual Environment (MassIVE) at https://massive.ucsd.edu under the identifier MSV000093148. Raw metagenomic data is publicly available under NCBI Bio Project ID PRJNA967692 and biosamples SAMN34718313-SAMN34718326. All data are available from the corresponding author upon request.

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

## Acknowledgements

We thank Linh Anh Cat and Lauren Pandori for their assistance with access to the Cabrillo State Marine Reserve (Permit #: CABR-2023-SCI-0006), Brendan M. Duggan (Skaggs School of Pharmacy, UC San Diego) for access to NMR equipment and expert guidance, Douglas A. Sweeney, Alyssa M. Demko, and Leesa J. Klau for their contributions to the early development of SMIRC, and Roger Linington for the Tanimoto similarity coefficient calculated for cabrillospiral A. This research was supported by the National Institutes of Health grants R21AT010493 (P.R.J., T.F.M.), R01GM085770 (P.R.J.), RO1CA184984 (L.F.W., S.J.A.), and R21AI171824/R01AI158612 (A.J.O.).

## Author contributions

Research design: A.B., A.B.C., T.F.M., P.R.J. Methodology: A.B., M.N.S., A.B.C., H.H., M.N.M., S.L., E.B. da S. Data analysis: A.B., M.N.S., A.B.C., H.H., S.L., A.J.O., L.F.W., S.J.A., T.F.M., P.R.J. Writing: A.B., A.B.C., H.H., T.F.M., P.R.J. with contributions from all authors.

## Competing interests

The authors declare no competing interests.
