## [Peer Review File · Nature Communications]

Small Molecule In Situ Resin Capture Provides a Compound First Approach to Natural Product DiscoveryREVIEWER COMMENTS

Reviewer #1 (Remarks to the Author):

The manuscript describes an interesting method that captures natural products directly from marine environments at amounts that can be suitable for NMR-based structure elucidation. This was demonstrated with several technical setups and habitats and resulted in two series of new polyketides, named cabrillostatin and the cabrillospirals, in addition to some known natural products. While ample work exists on natural product discovery from extracted individual organisms, knowledge on natural roles of metabolites and their occurrence in complex communities is scarce. As a method that can be readily implemented in research labs, "SMIRC" could facilitate such studies. In addition to the methodology, the manuscript features some impressive structure elucidation work from small substance amounts as well as metagenomic studies that attempted to identify compound sources. The latter were unfortunately not successful and the section might be shortened and parts of it moved to the SI.

Further comments/questions:

- 1) Abstract: "Isolated two novel carbon skeletons" should be rephrased to "isolate compounds with two novel carbon skeletons" or similar.
- 2) Page 3: "Despite this realization, our best discovery efforts have failed to access this predicted novelty...": This should be rephrased. It reads as if nobody was able to obtain novel natural products based on bioinformatic predictions.
- 3) Page 10: Could the vinylogous formate ester be an artifact arising from, e.g., dehydration of an unstable and structurally less unusual cyclic hemiacetal? This might explain why no activity was found. Did the authors detect a candidate for a congener by LC-MS?
- 4) Page 14: ...and complemented by...
- 5) Page 18: Explain the specialized term "hybrid KS".

6) Page 7: It seems unlikely that the cabrillospiral PKS would contain >10 almost identical KS domains. Can the authors explain the rationale?

Reviewer #2 (Remarks to the Author):

The manuscript by Bogdanov and colleagues describes a culture-independent method to discover novel natural products based on the deployment of adsorbent resin in the environment to capture natural product in situ. While previous work from this group already described deployment of resin in the environment to study microbial products (<https://doi.org/10.1128/AEM.02830-18>), it was thus far followed only by metabolomics paired with metagenomics. Here the authors add on top of these approaches the isolation of purified molecules and their characterization by NMR, MS and in terms of bioactivity. Notably, the authors describe the discovery of two new carbon skeletons. This achievement demonstrates that it is in fact possible to obtain sufficient amount of purified compounds directly from the environment for structural elucidation by NMR which is impressive and of great interest to the natural products community. In addition to offering a way to address questions in chemical ecology, as a culture-independent approach it should also provide access to molecules that are not produced by microorganisms that are readily cultured or are not expressed in standard laboratory conditions.

According to the authors, this paper describes a method but the potential future users among the readers will regret the lack of a critical analysis by the authors of the limitations of the approach. A lot of attention is brought to the validation of particular abilities (structure resolution, isolation of bioactive molecules, linking molecules to their producing gene clusters) with various degrees of success but it remains hard to forge an opinion on the scalability of this approach as a viable new paradigm in natural product discovery. The authors should address the key question of scalability by sharing the main bottleneck of the approach, according to them, and provide insights from their experience regarding its limitations.

While some minor variations are introduced (embedding the resin in agar) their impact does not seem clear as the most successful part of the work, which led to the discovery of two

new carbon skeletons does not make use of it and appears identical in terms of methodology to the previous use of this approach by this group (<https://doi.org/10.1128/AEM.02830-18>). What enabled this success for these molecules? Was it to attempt the approach in multiple sites to find one of suitable chemical complexity? Was it a question of scale? If so, would using more resin be desirable and practical? Providing such answers and setting clear expectations in terms of practical difficulty in the main text will most likely impact a lot the potential for use of the approach by the community.

Specific comments:

A brief explanation of why HP-20 is the best resin to use for SMIRC would be nice, would there be interesting alternatives?

Method section could indicate what type of precision scale and method was used to precisely weight such low amount of material (a few dozens of μg). It would be good to state the error range expected since this could impact a lot the concentrations reported for the bioactivities.

It is a bit confusing that the chromatograms in Figure 1 are UV with microfractionation bioassays while they are mass chromatogram without bioassay data in Figure 2. For a potential user of the approach, it would be easier to gain an understanding of what to expect from a typical SMIRC trace if the format was consistent.

DFT calculation was carried out, but the calculated data is not provided by the authors. Only experimental ECD spectra is shown in Fig. S11, calculated ECD spectra should be included to compare it with the experimental ones. Calculated chemical shifts and 3D Cartesian coordinates of the structures in Fig. S12, Fig. S13 and Fig. S14 should be provided as well.

The authors could provide more context around the claim of bioactivity of cabrillostatin. In its current format, the description of bioactivity data (line 203-231) makes it hard for a non-

expert to understand why this panel of assays was chosen in particular. Each assay reported here seem to lead to some signal for cabrillostatin, were more assays conducted but not reported ? If yes, including the full list would be of great interest to the readers wishing to attempt SMIRC in the future and the negative data on the other assays would make a stronger case for specific cabrillostatin activities. Currently, the formulation “intriguing biological activities” used by the authors line 372 is indeed the only conclusion a non-expert would reach and does not appear sufficient to “demonstrate the applications of this approach to natural product drug discovery » (line 374). To be clear, I do think there is potential to discover novel active compounds with SMIRC but unless this part is made clearer by the authors, I suggest removing “broad” in the abstract line 39, removing “promising” line 76, and adding the word “potential” line 374.

Metagenomics:

The metagenomic analyses part describes an ambitious attempt at identifying the cabrillostatin and cabrillospirals BGCs which failed to reach a solid conclusion but nevertheless should be applauded.

Since the demonstration of identifying the BGC is not achieved in this work, it is unclear in the abstract why metagenomic are “introduced” in this work line 36 since an attempt at metagenomics was already present in the group’s previous work with resin deployment in situ. Similarly, “link compounds to producing organisms » line 37 of the abstract is very speculative and should be removed unless a more solid demonstration is provided. Such overclaims will diminish the perceived novelty of a truly successful demonstration in the future by these authors or others.

The assembled sequences of the two candidate Type1PKS BGCs proposed by the authors are too small for the cabrillospirals, yet the authors showcase these BGCs and the associated planctomycte MAG as their top candidates because they consider that the assembled sequence they report is truncated due to assembly challenges commonly associated with highly repetitive modular PKSs. While this effect is indeed sometimes reported it is not a rule. In any case, this effect should not affect the rest of the contig which is not as repetitive

as the PKS and the decorating genes should be faithfully represented. Is there any indication to be found there? In a recent paper (<https://doi.org/10.1038/s41396-023-01410-3>), the same authors made the case of a BGC as the origin of an halogenated compound based on the presence of halogenase genes in it. Their absence on the proposed BGCs of a dihalogenated molecule seem to contradict the authors' conviction in their top candidate.

Reviewer #3 (Remarks to the Author):

In "Small Molecule In Situ Resin Capture – A Compound First Approach to Natural Product Discovery," Bogdanov et al. report on the natural product isolation and structure elucidation directly from the environment. Their approach makes use of a hydrophobic resin which is deployed in the environment to capture natural products directly where they are produced and act. They then use a combination of state-of-the-art mass spectrometry and NMR methods to dereplicate and elucidate a range of compounds, including so far unknown compounds with new carbon backbones.

Their approach tackles a central bottleneck in common culture-based NP discovery approaches, which is the difficulty of culturing microbes and the often encountered silence of the genetic potential in culturing conditions. While the approach of resin-based environmental sampling is not necessarily novel, the authors bring its capabilities for NP discovery, purification, and de novo structure elucidation to a new level, which I found quite impressive. In addition to solving new structures out of ultra-complex environmental samples, it is great to see that the authors used public MS/MS data and state-of-the-art computational metabolomics tools to compare their own data to other public datasets to check for the occurrence of their newly identified molecules in other marine environments, mainly from the Pacific.

The connection of environmental sampling, de novo structure elucidation, and repository scale data analysis, is quite innovative in my opinion and could be a prime example for future NP discovery studies. In my opinion, this paper should be of broad interest for both the natural product, marine chemical ecology, as well as the metabolomics community, and I think that Nature Communications would be a great place for it.

While I am in principle very enthusiastic about the paper, there are a couple of points the authors may want to consider and revise before the paper gets published.

Besides your own metagenome data, is there any evidence or hints to other ocean metagenomes and putative gene clusters thereof, that could be linked to the biosynthesis of cabrillostatins?

For cabrillospirals, are Planctomycetes accessible by current culturing methods? If yes, did you attempt to cultivate and verify the production from Planctomycetes?

As for the comparison of your data against the repository, I am wondering to what degree ecological insights can be drawn from these dataset matches?

Other specific comments:

Line 61: Can you provide some examples/numbers for the richness of the Earth's microbiome? Perhaps from the Earth Microbiome Project or other initiatives?

Line 125: How likely is it that the compounds discovered are made by the bacteria growing on the resin? Did you do any cultivation and MS experiments of the pink colony-forming bacteria?

Line 147: Why did you target specifically this compound?

Line 152: What is "tandem UV"? Or do you mean coupled UV and ELSD detection?

Line 191: What is the reason/bottleneck to assign the stereochemistry? Will this be feasible with the amounts of compound you isolate? Or is this a limitation of the approach? As you stated, it sounds a little vague. Especially as the paper has been published a while ago, I would assume you could estimate whether this will be feasible by now.

Line 250: You describe the identification of a wide range of compounds using MS/MS matching. What confidence level (according to the Sumner et al. levels) were these? Did you

confirm any of them with authentic standards?

Line 436: Why did you use a 4.6 mm column (semi-prep scale) at such a high flow rate?

Typically, a 2 mm column and lower flow rates are more appropriate for ESI, or did you do flow splitting?

Line 449: Please provide the GNPS settings.

SI Fig. S3: Change the title to "HR-ESIMS/MS".

We thank the reviewers for their helpful comments, which we have addressed point by point below. Our responses are in italics.

REVIEWER COMMENTS

Reviewer #1 (Remarks to the Author):

The manuscript describes an interesting method that captures natural products directly from marine environments at amounts that can be suitable for NMR-based structure elucidation. This was demonstrated with several technical setups and habitats and resulted in two series of new polyketides, named cabrillostatin and the cabrillospirals, in addition to some known natural products. While ample work exists on natural product discovery from extracted individual organisms, knowledge on natural roles of metabolites and their occurrence in complex communities is scarce. As a method that can be readily implemented in research labs, “SMIRC” could facilitate such studies. In addition to the methodology, the manuscript features some impressive structure elucidation work from small substance amounts as well as metagenomic studies that attempted to identify compound sources. The latter were unfortunately not successful and the section might be shortened and parts of it moved to the SI.

Response: We reduced the metagenomic section of the paper without the need to move any text to the SI.

Further comments/questions:

1) Abstract: “Isolated two novel carbon skeletons” should be rephrased to “isolate compounds with two novel carbon skeletons” or similar.

Response: change made as suggested.

2) Page 3: “Despite this realization, our best discovery efforts have failed to access this predicted novelty...”: This should be rephrased. It reads as if nobody was able to obtain novel natural products based on bioinformatic predictions.

Response: the text has been changed to “Despite this realization, our best discovery efforts have failed to access the vast majority of this predicted chemical space and thus new approaches to natural product discovery are needed.”

3) Page 10: Could the vinylogous formate ester be an artifact arising from, e.g., dehydration of an unstable and structurally less unusual cyclic hemiacetal? This might explain why no activity was found. Did the authors detect a candidate for a congener by LC-MS?

Response: As for many natural products, it's possible that the compound isolated (2-3, vinylogous formate esters) are artifacts of subsequent chemical (or enzymatic) reactions that occur outside the producing organism. However, a detailed search of the LCMS data did not find any candidate congeners that would support this possibility. We also did not see a methoxy analog of the hemiacetal, which might be expected given the use of methanol in the extraction. We added text to clarify this point. Even if 2-3 are not natural products, the novelty of the structures is not diminished. Finding the associated BGC could help resolve this question.

4) Page 14: ...and complemented by...

Response: change made as suggested

5) Page 18: Explain the specialized term “hybrid KS”.

Response: we have spelled out adenylation (A) and ketosynthase (KS) domains in this sentence as it is their first use and revised the text to clarify that it is a hybrid PKS-NRPS BGC that was predicted.

6) Page 7: It seems unlikely that the cabrillospirals PKS would contain >10 almost identical KS domains. Can the authors explain the rationale?

Response: We have removed this text in the process of shortening this section in response to a prior comment.

Reviewer #2 (Remarks to the Author):

The manuscript by Bogdanov and colleagues describes a culture-independent method to discover novel natural products based on the deployment of adsorbent resin in the environment to capture natural product in situ. While previous work from this group already described deployment of resin in the environment to study microbial products (<https://doi.org/10.1128/AEM.02830-18>), it was thus far followed only by metabolomics paired with metagenomics. Here the authors add on top of these approaches the isolation of purified molecules and their characterization by NMR, MS and in terms of bioactivity. Notably, the authors describe the discovery of two new carbon skeletons. This achievement demonstrates that it is in fact possible to obtain sufficient amount of purified compounds directly from the environment for structural elucidation by NMR which is impressive and of great interest to the natural products community. In addition to offering a way to address questions in chemical ecology, as a culture-independent approach it should also provide access to molecules that are not produced by microorganisms that are readily cultured or are not expressed in standard laboratory conditions.

According to the authors, this paper describes a method but the potential future users among the readers will regret the lack of a critical analysis by the authors of the limitations of the approach. A lot of attention is brought to the validation of particular abilities (structure resolution, isolation of bioactive molecules, linking molecules to their producing gene clusters) with various degrees of success but it remains hard to forge an opinion on the scalability of this approach as a viable new paradigm in natural product discovery. The authors should address the key question of scalability by sharing the main bottleneck of the approach, according to them, and provide insights from their experience regarding its limitations.

Response: We have expanded on our analysis of the limitation of the approach throughout the text, which includes the following:

“although challenges in these areas remain”.

“Notably, compounds 1, 2, and unidentified compound m/z 757.4630 $[M+H]^+$ were consistently detected in three separate deployments spanning seven months, suggesting that repeat deployments can provide a path forward to overcome supply bottlenecks for some target compounds (Fig. S11).”

“While our efforts to link these compounds to their biological origins remain ongoing, the challenges associated with linking compounds with gene clusters in complex systems such as the ones sampled here remain clear.”

“Yet supply issues remain the major bottleneck with this approach, which in the case of cabrillostatin should be resolvable via synthetic methods. More complex structures such as the cabrillospirals will require a larger investment from the synthetic community or beyond to meet supply challenge. Ultimately, compound supply will need to be addressed on a case-by-case basis and the upper limits of scale-up

deployments vetted. Regardless of the approaches taken to address resupply challenges, the discovery of new natural product chemotypes remains a top priority for natural products research and a driving force in drug discovery.”

“While our efforts to link these compounds to their biological origins remain ongoing, it is clear that establishing these types of linkages in complex systems such as the ones sampled here remain a major challenge.”

While some minor variations are introduced (embedding the resin in agar) their impact does not seem clear as the most successful part of the work, which led to the discovery of two new carbon skeletons does not make use of it and appears identical in terms of methodology to the previous use of this approach by this group (<https://doi.org/10.1128/AEM.02830-18>).

Response: While embedding the resin in agar did not lead to the discovery of a new compound, it had a dramatic effect on the chemical profile of the extracts (Figure 1) and yielded a known compound that was not observed when resin was used alone. We think this is a particularly important modification to the technique that warrants further study. We did not use the agar modification in the initial AEM paper cited in this comment. The agar modification was new to the present study.

What enabled this success for these molecules? Was it to attempt the approach in multiple sites to find one of suitable chemical complexity? Was it a question of scale? If so, would using more resin be desirable and practical? Providing such answers and setting clear expectations in terms of practical difficulty in the main text will most likely impact a lot the potential for use of the approach by the community.

Response: We deployed the resins at 4 sites and the resulting metabolomes were very different from each other (Fig. 5A). It's not clear why the CSMR was more productive. We revised the text to address this point: “While determining what drives these differences, or why the CSMR extracts appeared enriched in novel compounds, will require more study, these observations support deployments in diverse habitats to further explore the discovery potential of the resin capture approach.” We also added a sentence to the discussion: “Why this site appeared particularly rich in novel compounds remains unknown, but the potential benefits of increased biodiversity within a marine protected area may have played a role.” The scale of the deployments can certainly play a role and is something that needs to be considered for initial deployments. We have added a sentence to address this: “Yet others were not, indicating that temporal variability is an important consideration when selecting the quantity of resin to deploy.”

Specific comments:

A brief explanation of why HP-20 is the best resin to use for SMIRC would be nice, would there be interesting alternatives?

Response: We selected HP20 based on literature reports that it was effective and superior to other adsorbent resins. We modified the following sentence and added a reference to address this comment: “SMIRC employs the adsorbent resin HP-20, which has proven effective for the adsorption of lipophilic marine toxins²⁶, to capture these small organic molecules after they are released and thus provides a mechanism to access environmental metabolomes.”

Method section could indicate what type of precision scale and method was used to precisely weight such low amount of material (a few dozens of µg). It would be good to state the error range expected since this could impact a lot the concentrations reported for the bioactivities.

Response: We used the ¹³C-satellite method to quantify the sub-milligram amounts. The precision of this quantitative NMR method is related to the electronic integration of the NMR signals, which is typically 5-10% (ref 72) and has been added to the text. While this error was unlikely to have affected the results, we have nonetheless added to the text that special quantification methods are required to determine test concentrations.

It is a bit confusing that the chromatograms in Figure 1 are UV with microfractionation bioassays while they are mass chromatogram without bioassay data in Figure 2. For a potential user of the approach, it would be easier to gain an understanding of what to expect from a typical SMIRC trace if the format was consistent.

Response: Figure 1 describes the bioassay guided isolation of an active compound. Figure 2 describes extracts from the CSMR site that were not active, so we instead used the MS data to guide the isolation of new compounds. We have added text to better clarify this. Figure S1 provides a typical UV trace and MS chromatogram from a crude extract. We have modified the legend to indicate that this is a typical extract and that the lack of resolution reflects the complexity of the extract. We have also added the following sentence to better inform users of what to expect. "This level of complexity was common to the crude extracts, which required extensive purification."

DFT calculation was carried out, but the calculated data is not provided by the authors. Only experimental ECD spectra is shown in Fig. S11, calculated ECD spectra should be included to compare it with the experimental ones. Calculated chemical shifts and 3D Cartesian coordinates of the structures in Fig. S12, Fig. S13 and Fig. S14 should be provided as well.

Response: We expanded the DFT calculations for the structures shown in the Figures S12-S14 and calculated the ECD spectra for 5S,6S, 5S,6R, 5R,6S and 5R,6R diastereomers. We established the absolute configuration of the A ring in cabrillospirals A and B based on the comparison of the experimental and calculated ECD spectra. We also corrected the configuration putatively assigned to cabrillospiral B (changed Figure 3 and added text to the main manuscript page 10, paragraph 2 and to the structure elucidation part of supplementary information). We added a new figure (Fig. S10) with the calculated ECD spectra and updated figures S7-S9. We included tables with 3D cartesian coordinates to the supplementary information (Tables S14-S18).

We have assigned the absolute configuration of a substantial part of molecules 2 and 3. The complexities of the final and complete stereochemical determinations, including DFT calculated NMR chemical shifts for the whole molecule, demand deeper and more sophisticated investigations that we are beyond the scope of this Communications paper.

The authors could provide more context around the claim of bioactivity of cabrillostatin. In its current format, the description of bioactivity data (line 203-231) makes it hard for a non-expert to understand why this panel of assays was chosen in particular. Each assay reported here seem to lead to some signal for cabrillostatin, were more assays conducted but not reported? If yes, including the full list would be of great interest to the readers wishing to attempt SMIRC in the future and the negative data on the other assays would make a stronger case for specific cabrillostatin activities.

Response: Our choice of assays for cabrillostatin was first driven by reports that the amino acid statine was a phamacophore for protease activity. Here we report results from all proteases tested including those with no activity. To obtain additional bioactivity data, we employed cell painting and high content imaging using cancer cells in a manner that would be more informative than simple cytotoxicity assays. We reasoned that testing a diverse set of cancer cell lines using this approach would improve the chances of capturing cellular responses and provide information about potential mechanisms of action. These assays demonstrated significant activity and some degree of selectivity. We have modified the following

text to better clarify this point: “Quantifying the separation showed that I was significantly bioactive ($p < 10^{-6}$) at 10 mM against six of the nine cell lines (Fig. 4B, Table S7) but not against OVCAR4, HepG2, and 786-O cells indicating a degree of selectivity.” The final assay employed an induced pluripotent stem cell-derived cardiomyocyte (iPSC-CM) model. This offered a readout of cell functionality instead of biomarkers, which provided a complementing assay to the cancer cell lines phenotypic profiling. It’s correct that cabrillostatin displayed activity in all of the assays (but not against all proteases or cell lines), raising the interesting question as to why this biologically active compound appears to be so widely distributed.

Currently, the formulation “intriguing biological activities” used by the authors line 372 is indeed the only conclusion a non-expert would reach and does not appear sufficient to “demonstrate the applications of this approach to natural product drug discovery » (line 374). To be clear, I do think there is potential to discover novel active compounds with SMIRC but unless this part is made clearer by the authors, I suggest removing “broad” in the abstract line 39, removing “promising” line 76, and adding the word “potential” line 374.

Response: Changes made as suggested.

Metagenomics:

The metagenomic analyses part describes an ambitious attempt at identifying the cabrillostatin and cabrillospirals BGCs which failed to reach a solid conclusion but nevertheless should be applauded. Since the demonstration of identifying the BGC is not achieved in this work, it is unclear in the abstract why metagenomic are “introduced” in this work line 36 since an attempt at metagenomics was already present in the group’s previous work with resin deployment in situ. Similarly, “link compounds to producing organisms » line 37 of the abstract is very speculative and should be removed unless a more solid demonstration is provided. Such overclaims will diminish the perceived novelty of a truly successful demonstration in the future by these authors or others.

Response: We appreciate this comment and have revised the relevant sentence in the abstract as follows: “Expanded deployments, in situ cultivation, and metagenomics were employed to facilitate compound discovery, enhance yields, and attempt to link compounds to producing organisms, although the complexity of the microbial communities created challenges for the later.” We have also modified the discussion to better address this comment: “While efforts to link compounds to producers have been successful in the context of symbioses^{33,65}, the results presented here reinforce the challenges associated with linking compounds to producers in complex microbial communities.”

The assembled sequences of the two candidate Type I PKS BGCs proposed by the authors are too small for the cabrillospirals, yet the authors showcase these BGCs and the associated planctomycte MAG as their top candidates because they consider that the assembled sequence they report is truncated due to assembly challenges commonly associated with highly repetitive modular PKSs. While this effect is indeed sometimes reported it is not a rule. In any case, this effect should not affect the rest of the contig which is not as repetitive as the PKS and the decorating genes should be faithfully represented. Is there any indication to be found there? In a recent paper (<https://doi.org/10.1038/s41396-023-01410-3>), the same authors made the case of a BGC as the origin of an halogenated compound based on the presence of halogenase genes in it. Their absence on the proposed BGCs of a dihalogenated molecule seem to contradict the authors’ conviction in their top candidate.

Response: We have reduced this section of the paper in response to a comment by reviewer 1 and to address the concerns raised here. This includes the following modified text: “While lacking key biosynthetic genes, the planctomycte MAG was nonetheless the best candidate identified.”

Reviewer #3 (Remarks to the Author):

In “Small Molecule In Situ Resin Capture – A Compound First Approach to Natural Product Discovery,” Bogdanov et al. report on the natural product isolation and structure elucidation directly from the environment. Their approach makes use of a hydrophobic resin which is deployed in the environment to capture natural products directly where they are produced and act. They then use a combination of state-of-the-art mass spectrometry and NMR methods to dereplicate and elucidate a range of compounds, including so far unknown compounds with new carbon backbones.

Their approach tackles a central bottleneck in common culture-based NP discovery approaches, which is the difficulty of culturing microbes and the often encountered silence of the genetic potential in culturing conditions. While the approach of resin-based environmental sampling is not necessarily novel, the authors bring its capabilities for NP discovery, purification, and de novo structure elucidation to a new level, which I found quite impressive. In addition to solving new structures out of ultra-complex environmental samples, it is great to see that the authors used public MS/MS data and state-of-the-art computational metabolomics tools to compare their own data to other public datasets to check for the occurrence of their newly identified molecules in other marine environments, mainly from the Pacific.

The connection of environmental sampling, de novo structure elucidation, and repository scale data analysis, is quite innovative in my opinion and could be a prime example for future NP discovery studies. In my opinion, this paper should be of broad interest for both the natural product, marine chemical ecology, as well as the metabolomics community, and I think that Nature Communications would be a great place for it.

While I am in principle very enthusiastic about the paper, there are a couple of points the authors may want to consider and revise before the paper gets published.

Besides your own metagenome data, is there any evidence or hints to other ocean metagenomes and putative gene clusters thereof, that could be linked to the biosynthesis of cabrillostatins?

Response: This is a great question. Our best biosynthetic hook for cabrillostain is the hybrid KS predicted to extend leucine to yield the non-proteogenic amino acid statine. In response to this question, we blasted the two reference KSs linked to statine biosynthesis against an extensive KS library derived from metagenomes, and we found no matches. We have added the following text: “Interestingly, the two reference KSs shared <60% amino acid identity with a database of metagenome-extracted KSs⁵⁹, supporting the rarity of statine incorporation into natural products.

For cabrillospirals, are Planctomycetes accessible by current culturing methods? If yes, did you attempt to cultivate and verify the production from Planctomycetes?

Response: We tried to cultivate Planctomycetes from the resin and from seagrass and algae samples collected at the site but these efforts were not successful. We have added mention of this to the text.

As for the comparison of your data against the repository, I am wondering to what degree ecological insights can be drawn from these dataset matches?

Response: While few ecological insights can be drawn from our study, this approach will hopefully inspire future studies in chemical ecology. E.g., the detection of antimicrobial seagrass flavonoids in seawater suggests these compounds could shape community structure in seagrass beds. It’s also intriguing that the biologically active cabrillostatin and related compounds were commonly observed in seawater metabolome datasets.

Other specific comments:

Line 61: Can you provide some examples/numbers for the richness of the Earth's microbiome? Perhaps from the Earth Microbiome Project or other initiatives?

Response: An estimate for the richness of Earth's microbiome and the associated referene has been added.

Line 125: How likely is it that the compounds discovered are made by the bacteria growing on the resin? Did you do any cultivation and MS experiments of the pink colony-forming bacteria?

Response: It can't be ruled out that bacteria growing on the resin are a source of some of the compounds captured. This concept is what inspired us to embed the resin in agar to encourage "in situ microbial growth". Our efforts to obtain compound producers in pure culture have not been successful to date. We added this information to the text.

Line 147: Why did you target specifically this compound?

Response: Cabrillosstatin was targeted because it appeared to be novel and had a relatively high-intensity molecular peak, which suggested it could be recoverable. We modified the text accordingly.

Line 152: What is "tandem UV"? Or do you mean coupled UV and ELSD detection?

Response: Yes, we meant coupled UV and ELSD detection and have changed the text accordingly.

Line 191: What is the reason/bottleneck to assign the stereochemistry? Will this be feasible with the amounts of compound you isolate? Or is this a limitation of the approach? As you stated, it sounds a little vague. Especially as the paper has been published a while ago, I would assume you could estimate whether this will be feasible by now.

Response: Determination of the absolute configuration of cabrillospirals A/B is independent of compound availability and cannot be done with spectroscopic methods alone. Given its three independent stereoclusters and a high degree of free rotation, a partial or total chemical synthesis will likely be required. We have modified the text to clarify this: "The full assignment of the relative and absolute configurations remains challenging and will likely require synthetic efforts."

Line 250: You describe the identification of a wide range of compounds using MS/MS matching. What confidence level (according to the Sumner et al. levels) were these? Did you confirm any of them with authentic standards?

Response: We have added that these matches were level 2 along with the Sumner reference. None were made by comparison with authentic standards, which can prove extremely difficult to acquire.

Line 436: Why did you use a 4.6 mm column (semi-prep scale) at such a high flow rate? Typically, a 2 mm column and lower flow rates are more appropriate for ESI, or did you do flow splitting?

Response: While not ideal, the 4.6 mm column is our standard protocol (without flow splitting) and was used to provide a general overview of extract composition and to guide compound isolation. However, we have added to the Methods section that a smaller column was used for the feature-based analyses.

Line 449: Please provide the GNPS settings.

Response: The settings are provided in detail in the GNPS documentation as cited. Given the length of the Methods section, we have elected not to repeat them here. We have clarified in the text where the settings can be found. “Classic Molecular networking and automated library searches were performed using the online workflow and the settings detailed in the GNPS documentation (<https://ccms-ucsd.github.io/GNPSDocumentation/>)”.

SI Fig. S3: Change the title to “HR-ESIMS/MS”.

Response: Change made as suggested.

REVIEWERS' COMMENTS

Reviewer #2 (Remarks to the Author):

The authors have addressed the points I raised in my initial review. The new text provides the readers with more details regarding the limitations of the approach. I believe this work shows the potential of in situ resin capture and this paper will be a good guide for researchers interested in replicating the strategy.

Reviewer #3 (Remarks to the Author):

I would like to thank the authors for addressing my comments and in principle I would be happy to endorse the publication of their manuscript.

The only thing I would suggest them to still modify before the paper gets published is the reference to the GNPS settings. As the authors are probably well aware, settings such as mass tolerance, cosine cut-off and filtering options are user defined and have to be selected for each networking job. Hence, referring to the general documentation of GNPS is not providing these information.

Response to REVIEWERS' COMMENTS

Reviewer #2 (Remarks to the Author):

The authors have addressed the points I raised in my initial review. The new text provides the readers with more details regarding the limitations of the approach. I believe this work shows the potential of in situ resin capture and this paper will be a good guide for researchers interested in replicating the strategy.

Reviewer #3 (Remarks to the Author):

I would like to thank the authors for addressing my comments and in principle I would be happy to endorse the publication of their manuscript.

The only thing I would suggest them to still modify before the paper gets published is the reference to the GNPS settings. As the authors are probably well aware, settings such as mass tolerance, cosine cut-off and filtering options are user defined and have to be selected for each networking job. Hence, referring to the general documentation of GNPS is not providing these information.

Response: the GNPS settings have been added to the main manuscript as suggested.